# Improving Active-Learning Evaluation, with Applications to Protein-Property Prediction

## Abstract

We highlight that current evaluations of active-learning methods often fail to reflect important aspects of real-world applications, giving an incomplete picture of how methods can behave in practice. Most notably, evaluation problems are commonly constructed from heavily curated datasets, limiting their ability to rigorously stress-test data acquisition: even the worst acquirable data in these datasets is often reasonably useful with respect to the task at hand. To address this we introduce Active Learning on Protein Sequences (`ALPS`), a set of problems constructed to test key challenges that active-learning methods need to handle in real-world settings. We use `ALPS` to assess a number of previously successful methods, revealing a number of interesting behaviours and methodological issues. The `ALPS` codebase serves to support straightforward extensions of our evaluations in future work.

## 1 Introduction

Active learning involves seeking the best data for training a model; typically this means adaptively choosing inputs to acquire labels for (Atlas et al, 1989; Settles, 2012). Empirical evaluations have helped show the benefit of intelligent data acquisition, with several successful demonstrations in recent years (Bickford Smith et al, 2023; 2024; Hübotter et al, 2024; 2025; Melo et al, 2024; Osband et al, 2023). But we argue that existing evaluations often fail to reflect key challenges in practical applications, limiting our ability to gauge how methods will really perform beyond academic studies.

The principal issue we highlight is the use of heavily curated datasets in the construction of active-learning problems. It is common for example to use standard academic datasets from computer vision (Bengar et al, 2021; Chan et al, 2021; Lüth et al, 2023; Mittal et al, 2019; 2023; Siméoni et al, 2020) and natural-language processing (Ein-Dor et al, 2020; Maekawa et al, 2022; Margatina et al, 2022; Seo et al, 2022; Yuan et al, 2020), with typical curation steps including ensuring a roughly equal number of examples per class and removing unrepresentative examples. By using these curated data sources in place of the messy ones often used in the real world, existing evaluations give us a false sense of the active-learning methods we are assessing. If all acquirable data has already been filtered to be at least moderately useful for the task at hand, there is an artificial limit on how badly any method can perform, harming our ability to detect weaknesses in methods. On top of this, even if evaluations emphasise the cost of acquiring labels, they crucially hide the cost of implicit curation steps, leading us to overestimate the real-world performance achievable for a given cost.

We therefore believe there is a critical need to complement existing active-learning problems with new ones that reflect underrepresented challenges. We suggest a promising context within which to design new problems is protein-property prediction, namely the task of mapping from a protein's sequence of amino acids to some measure of its behaviour (Lesk, 2019). One reason for this is the scope for concrete impact: better protein-property prediction could enable advances in practical pursuits like protein engineering as well as foundational research in biology (Notin et al, 2023; 2024). Another is that labelling protein sequences usually requires costly lab experiments, meaning there is much less labelled data available than in domains like computer vision and natural-language processing, and there is an ongoing pressing need for acquiring informative new labels. Meanwhile the labelled data that *is* currently available, thanks to past investments in experimental data-gathering (Bryant et al, 2021; Faure et al, 2022; 2024; Johnston et al, 2024; Poelwijk et al, 2019; Pokusaeva et al, 2019; Wu et al, 2016), is sufficient to construct useful problems for foundational methods development. We thus have a basis for iteratively working towards larger quantities of high-quality labelled data.

Capitalising on this opportunity, we introduce Active Learning on Protein Sequences (`ALPS`), a set of problems derived from existing protein datasets. In five core problems we do as little as possible to constrain the data that can be acquired, with two of the problems having near-exhaustive coverage of a region of the input space. Nine additional problems extend from these core problems to pose further challenges for active-learning methods, including working with skewed label distributions, acquiring data under experimental restrictions, and dealing with large quantities of redundant inputs.

Putting `ALPS` to use, we experimentally investigate the performance of a number of active-learning methods that have seen success in existing evaluations. We find that `ALPS` reveals failure cases in these methods that have been underrepresented in past work, including miscalibration of predictive uncertainty, sensitivity to class imbalance, and unreliable scaling with increasing acquisition batch size. Given this, our work brings to light not only key issues in the design of active-learning evaluations but also priorities for future method development, with a particular need for more robust data acquisition. To accelerate progress along these lines, we provide an open codebase (`anonymous.4open.science/r/alps-95A3`) designed for flexible experimentation.

## 2  EVALUATING ACTIVE LEARNING

Our aim in this work is to improve the way we evaluate active-learning methods. We begin by establishing a clear sense of our brief as evaluators, with a focus on expected downstream utility.

**Setup**  Active learning can be broadly defined as the process of training a predictive model on data acquired by an adaptive policy, whose decisions depend on the model being trained (Atlas et al, 1989; MacKay, 1992). These decisions can take many forms, including choosing state transitions to observe in an environment (Mehta et al, 2022) or a subset of examples from a labelled dataset (Mindermann et al, 2022), but the most commonly studied setting—and the one we focus on here—is choosing unlabelled inputs to acquire labels for (Settles, 2012). Specifically we consider pool-based active learning (Lewis & Gale, 1994) of a model, $p_\phi(y|x)$, that maps inputs $x \in \mathcal{X}$ to labels $y \in \mathcal{Y}$: we have access to a pool of $n$ unlabelled inputs, $\mathcal{X}_{\text{pool}} \subseteq \mathcal{X}$, but we can only afford to acquire $m < n$ labels due to the cost of labeling, which we assume follows some distribution $y \sim p_{\text{train}}(y|x)$.

Pool-based active learning is typically broken down into a sequence of steps, $t \in (1, 2, \ldots, T)$, where each step comprises three substeps. First, the data-acquisition algorithm selects a batch of $b$ query inputs, $\boldsymbol{x}_t = (x_{t,i})_{i=1}^b$, where $x_{t,i} \in \mathcal{X}_{\text{pool}}$, often by maximising an acquisition function that estimates some notion of data utility. Second, the algorithm obtains labels, $\boldsymbol{y}_t$, where $y_{t,i} \sim p_{\text{train}}(y_{t,i}|x_{t,i})$, and adds $(\boldsymbol{x}_t, \boldsymbol{y}_t)$ to the training dataset, $\mathcal{D}_{\text{train}}$. Third, the model, $p_\phi(y|x)$, is updated on $\mathcal{D}_{\text{train}}$.

**Goal**  Evaluating active-learning methods requires a clear sense of what we want to achieve with them. A technically precise way to describe this is in terms of downstream utility or loss (von Neumann & Morgenstern, 1947). In machine learning we often evaluate trained predictive models, $f_n = f(\cdot; x_{1:n}, y_{1:n})$, using a form of frequentist risk (Berger, 1985), $R = \mathbb{E}_{p_{\text{eval}}(x_*, y_*)}[\ell(x_*, y_*, f_n(x_*))]$, where $p_{\text{eval}}$ denotes a reference system used as a source of ground truth and $\ell$ denotes a loss function. Standard evaluation metrics can be understood as estimators of the risk for particular choices of loss function (e.g. the misclassification rate arises from the zero-one loss). Reduced risk is therefore a concrete and well-established notion of what we could gain from intelligent data acquisition.

**Problem design**  As well as making it clear what we should measure in evaluations, writing down this formal goal highlights the many factors that control the dependence between an active-learning method and its performance, factors that we need to consider when designing problems. Among these are the predictive task, $\mathcal{X} \times \mathcal{Y}$; the loss, $\ell$; the pool, $\mathcal{X}_{\text{pool}}$; the label source, $p_{\text{train}}$; the reference system, $p_{\text{eval}}$; the machine-learning method, $f$; and the costs and budgets for compute and labels.

## 3  SHORTFALLS IN EXISTING EVALUATIONS

Next we discuss how existing active-learning problems do not allow us to fulfil our brief as evaluators. In particular we highlight issues that arise from using curated data and neglecting task adaptation.

**Using curated data**  A striking pattern across the literature is the use of standard academic datasets as a basis for constructing active-learning problems. We estimate (Appendix A) that 37% of recent active-learning evaluations use standard vision datasets (e.g. Caltech101, CIFAR-10, ImageNet,

MNIST), 9% use standard text datasets (e.g. 20 News-groups, CiteSeer, CORA, PubMed, Reuters), and 9% use standard UCI (Dua & Graff, 2017) datasets (e.g. Adult, Ionosphere, Iris, Wine). These datasets are often heavily preprocessed to allow easier model training, for example by ensuring a roughly equal number of examples per class and filtering out examples considered unrepresentative, irrelevant, or ambiguous (Aitchison, 2021; Krizhevsky, 2009; Russakovsky et al, 2014).

This represents a major shortfall in existing evaluations. If all the acquirable data has already been vetted for quality, then the difference between the best data and the worst data is small, limiting our ability to properly stress-test methods and leading us to overestimate possible real-world performance. For example, while BALD (Houlsby et al, 2011) has been shown to target obscure data (Bickford Smith et al, 2023), with potentially disastrous consequences for working with the uncurated data pools often encountered in practice (Ardila et al, 2020; Mahajan et al, 2018; Raffel et al, 2020), this failure mode is masked in evaluations based on curated data.

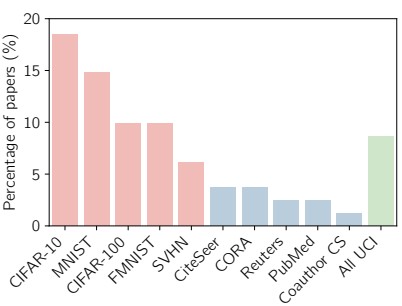

**Figure 1** Current evaluations of active learning rely heavily on standard vision, text and UCI datasets. Percentages here were estimated by taking recent papers from AISTATS, ICML, NeurIPS and UAI, filtering by active-learning keywords, randomly subsampling, manually discarding false positives (giving 81 papers), then listing the datasets used in empirical evaluations.

A top priority for new evaluations is therefore to use less heavily curated data. One way this might manifest is through the unlabelled pool: rather than only containing inputs that are likely to lead to useful labels (with respect to model performance), we should consider messier pools that include inputs that are unlikely to be useful, perhaps even comprising all inputs that could be labelled.

**Neglecting task adaptation**   Another key shortfall in current evaluations is failing to assess methods' abilities to adapt learning towards a particular task. In real-world applications we cannot expect our pool of unlabelled data to be tailored to the task of interest. For instance, we might want to predict whether a protein has a desired level of binding affinity with a target molecule, but the proportion of proteins in our pool that achieve this level might be very small and dependent on which molecule we are targeting. Yet existing evaluations tend to have a straightforward relationship between the active-learning problem and the source dataset from which it is constructed, such that all inputs relate to the task of interest, for example by exclusively belonging to the classes that occur at test time.

Given that a key motivation for active learning is the need to enhance a model for a particular task (Baumann et al, 2024; Bickford Smith et al, 2023; Hübotter et al, 2024; 2025; Osband et al, 2023; Tamkin et al, 2022), this common failure to consider task adaptation in evaluations is problematic. Like the use of curated data, it hinders our ability to rigorously test active-learning methods. A method with no notion of the task of interest is suboptimal in the general case, but the extent to which that manifests in evaluations will be limited if all acquirable data is relevant to the task at hand.

An additional requirement for new evaluations should therefore be to test how well active-learning methods can be tailored towards different tasks. Out of the many ways to implement this, perhaps the simplest is to use unlabelled pools within which not all inputs directly relate to the task of interest.

## 4 PROTEIN-PROPERTY PREDICTION

With a sense of the challenges we want to reflect in our evaluations, we turn to the question of how to implement them. We argue that the domain of protein-property prediction provides a compelling setting for this, due to its potential applied impact and the protein data at our disposal.

**Task**   In protein-property prediction we take as input a sequence, $x \in \mathcal{A}^L$, where $\mathcal{A}$ is a set of amino acids and $L$ denotes length, and produce as output a prediction of a property (or property vector), $y \in \mathcal{Y}$, that describes the protein's behaviour in a system (Lesk, 2019). Properties we might want to predict include the protein's solubility, its stability under changing conditions (e.g. temperature), or its binding affinity with a target of interest (e.g. a small molecule). Prediction of $y \in \mathbb{R}$ is sometimes reframed as classification by splitting the real line into bins: we can for example use a single threshold, such as the property value of a reference protein, to produce binary classification (Notin et al, 2023).

**Applications**  Protein-property prediction can unlock great value both in direct practical applications and in foundational research (Notin et al, 2023; 2024). In protein engineering (e.g. in the context of drug design), predictions can be used within optimisation objectives or constraints (Yang et al, 2018), perhaps to ensure any chosen protein satisfies a particular solubility requirement. In basic research, predictions can be used to characterise phenomena like epistasis (Olson et al, 2014), where changes at multiple locations in a protein sequence have an interacting effect on the protein's behaviour.

**Data**  Recent years have seen significant efforts to experimentally characterise protein behaviour, yielding a notable increase in the amount of available data (Chevalier et al, 2017; Tsuboyama et al, 2023). Commonly a reference protein (often a naturally occurring "wild-type" protein) and a number of variant proteins (with amino-acid sequences mutated from that of the reference protein) are synthesised and observed in some system, leading to a behaviour measurement for each of the proteins (Faure et al, 2024; Fowler & Fields, 2014; Kinney & McCandlish, 2019).

This source of data has four characteristics that are particularly relevant to our work. First, labels are expensive (Wittmann et al, 2022), justifying the use of careful data acquisition. Second, while there is now sufficient data available to construct interesting active-learning problems, there is still a pressing need for more data, and the space of possible new data is so vast that data-gathering needs to be targeted (e.g. with respect to promising candidates in protein design). Third, experimental datasets are typically not curated to the same extent as those often used in foundational machine-learning research (Wu et al, 2016; Johnston et al, 2024; Bryant et al, 2021; Notin et al, 2023; Pokusaeva et al, 2019). Fourth, labels are typically acquired in parallel; labelling proteins in batches of 96 is common, for example (Johnston et al, 2024; Pokusaeva et al, 2019; Yang et al, 2025).

We identify six existing datasets that we believe are particularly promising as a basis for constructing our new active-learning problems: AAV2 (Bryant et al, 2021), GB1 (Wu et al, 2016), GRB2 (Faure et al, 2024), His3 (Pokusaeva et al, 2019), mKate2 (Poelwijk et al, 2019), and TrpB (Johnston et al, 2024). Each dataset is named after a reference protein and contains measurements of the effect of mutating the reference protein, where the measurements correspond to the abundance—or, in the case of mKate2, the fluorescence intensity—of the protein variant after it is synthesised and subjected to a particular set of conditions. The datasets vary with respect to how the protein variants were selected: two datasets, TrpB and GB1, near-exhaustively enumerate $m$-amino-acid mutations for $m \leq 4$; the others cover greater degrees of mutation but are not close to being exhaustive.

## 5  ACTIVE LEARNING ON PROTEIN SEQUENCES

We now introduce Active Learning on Protein Sequences (ALPS), a set of new problems designed to help address the shortfalls identified in Section 3, building on the datasets discussed in Section 4.

### 5.1  THE ALPS PROBLEMS

**Uncurated data**  Our discussion in Section 3 stressed that the use of curated datasets can undermine evaluations by reducing the sensitivity of predictive performance to how data is acquired. We therefore start by designing five core problems, ALPS-Core-[AAV2,GB1,GRB2,mKate2,TrpB], in which we do as little as possible to constrain the data that can be acquired and simply aim to learn effective predictions for the whole dataset by setting $\mathcal{X}_{\text{pool}} = \mathcal{X}_{\text{test}}$. In real-world terms this translates to a scenario where we want to learn about the whole space of experimentally testable proteins and we can acquire a label for any protein in that space. As many common active-learning methods are designed for classification instead of regression, we binarise the labels using a wild-type protein's label as a threshold. These core problems are already a significant departure from the curated setups often used in existing active-learning evaluations: curation of the source data is minimal; two problems, ALPS-Core-GB1 and ALPS-Core-TrpB, near-exhaustively covering a region of input space.

Next we extend from these core problems to test the ability of active-learning methods to deal with skewed label distributions, which can be a practical challenge when working with uncurated data. We do this using ALPS-Unbalanced-TrpB-[2,5,12,17,23], five variants of ALPS-Core-TrpB with different degrees of class imbalance (ratio of class 0 to class 1) induced by varying the threshold used for binarising the labels. The different thresholds give rise to different decision boundaries.

**Task adaptation**    In Section 3 we argued for the importance of testing active-learning methods' ability to target a particular task when acquiring data. To this end, we introduce `ALPS-Redundant-His3`, which poses a scenario where the unlabelled pool contains a large number of redundant inputs that do not relate to the task of interest and cannot be labelled as class 0 or 1. More specifically we construct $\mathcal{X}_{\text{pool}}$ to contain inputs from classes 0 and 1 as well as inputs that belong to neither, while $\mathcal{X}_{\text{test}}$ has classes 0 and 1. If a redundant input is selected during acquisition, it is assigned to a third "neither" class; at test time the model only sees inputs from classes 0 and 1. Success on this problem requires identifying inputs that directly relate to the task of interest, namely classifying classes 0 and 1.

On top of this we design three more problems, `ALPS-Restricted-[GB1,GRB2,TrpB]`, that test an active-learning method's ability to gather data to aid predictions on inputs that we cannot acquire labels for, perhaps due to restrictions on what experiments can be run. Here $\mathcal{X}_{\text{pool}}$ contains only the inputs within $m$ mutations of the reference, while $\mathcal{X}_{\text{test}}$ contains only inputs with $m + 1$ mutations.

## 5.2  CODEBASE

We provide an open codebase implementing ALPS at `anonymous.4open.science/r/alps-95A3`. This codebase is rich with features that allow a wide range of experimentation with little effort. In particular, choices such as the type of embedding, acquisition strategy, and prediction head are all designed to be modular to allow easy isolated testing of different methodological components.

**Embeddings**    A key best practice in active learning is to capitalise on unlabelled data by using semi-supervised models, with a simple and generally applicable approach being to combine a fixed, unsupervised-pretrained encoder with a trainable, supervised prediction head (Bickford Smith et al, 2024). To support this we provide code for computing protein embeddings using 22 different pretrained encoders: 14 from the ESM family (Rives et al, 2021) and 8 from the ProtTrans family (Elnaggar et al, 2022). We additionally provide precomputed embeddings produced using the most advanced of these encoders, ESM3 (Hayes et al, 2025), for all of the ALPS problems. Notably the outputs of these encoders live in continuous spaces, which means we can use general-purpose prediction heads rather than models specialised to protein-property prediction.

**Prediction heads**    We provide code for a range of models (and corresponding learning algorithms): linear models; random forests; deterministic neural networks (with regularised maximum-likelihood training); Bayesian neural networks (with Laplace approximation, mean-field variational inference and Monte Carlo dropout); and Gaussian-process models (with variational inference).

**Acquisition methods**    We implement 12 data-acquisition methods in the ALPS codebase. Six use various measures of model uncertainty as a basis for acquisition: two fall within a Bayesian formulation (Rainforth et al, 2024), namely EPIG (Bickford Smith et al, 2023; 2024) and BALD (Houlsby et al, 2011), and the four others are predictive entropy (Settles & Craven, 2008), predictive margin (Scheffer et al, 2001), variation ratio (Gal, 2016) and mean standard deviation (Kendall et al, 2015). Four methods are based on notions of input- or feature-space coverage—greedy $k$ centres (Sener & Savarese, 2018), $k$ means (Pourahmadi et al, 2021), ProbCover (Yehuda et al, 2022) and TypiClust (Hacohen et al, 2022)—as well as BADGE (Ash et al, 2020) and BAIT (Ash et al, 2021). All uncertainty-based acquisition functions can be used for batch acquisition using the stochastic approach introduced by Kirsch et al (2023): the acquisition function is used to compute a distribution over batches of pool indices, then acquisition simply involves sampling from that distribution.

## 6  EXPERIMENTS

We now investigate how some popular active-learning methods deal with the ALPS problems. Given the vast array of possible setups that could be tested, this investigation is inevitably not exhaustive; its purpose is simply to demonstrate some of the insights that ALPS enables.

## 6.1  SETUP

**Model**    We primarily use an ESM3 encoder combined with a random-forest prediction head—random forests have supported effective data acquisition in past work (Bickford Smith et al, 2023; 2024; Kirsch, 2023; Kossen et al, 2021)—but also report some results with simpler biophysical (Georgiev, 2009) and onehot encoders, as well as linear and neural-network prediction heads.

| Encoder | Pred. head | Acq. method | NLL (250-500) | NLL (500-1000) | NLL (1000-2000) | Acc. (250-500) | Acc. (500-1000) | Acc. (1000-2000) |
|---|---|---|---|---|---|---|---|---|
| ESM3 | Linear model | Entropy | **0.067** | **0.065** | 0.056 | **0.984** | 0.987 | 0.991 |
| | | Random | 0.077 | 0.069 | 0.060 | 0.976 | 0.977 | 0.979 |
| | | TypiClust | 0.076 | 0.067 | 0.061 | 0.976 | 0.977 | 0.978 |
| | Neural network | BALD | 0.103 | 0.084 | 0.060 | 0.980 | 0.987 | 0.991 |
| | | EPIG | 0.104 | 0.086 | 0.052 | 0.983 | **0.987** | **0.992** |
| | | Random | 0.123 | 0.108 | 0.089 | 0.974 | 0.976 | 0.978 |
| | | TypiClust | 0.132 | 0.109 | 0.089 | 0.970 | 0.971 | 0.976 |
| | Random forest | BALD | 0.309 | 0.320 | 0.324 | 0.981 | 0.984 | 0.987 |
| | | EPIG | 0.104 | 0.111 | 0.114 | 0.980 | 0.982 | 0.985 |
| | | Random | 0.125 | 0.117 | 0.110 | 0.976 | 0.976 | 0.977 |
| | | TypiClust | 0.119 | 0.105 | 0.096 | 0.976 | 0.977 | 0.977 |
| Georgiev | Linear model | Entropy | 0.143 | 0.127 | 0.105 | 0.981 | 0.982 | 0.983 |
| | | Random | 0.106 | 0.096 | 0.079 | 0.974 | 0.974 | 0.976 |
| | | TypiClust | 0.114 | 0.113 | 0.105 | 0.975 | 0.974 | 0.974 |
| | Neural network | BALD | 0.118 | 0.075 | 0.054 | 0.981 | 0.986 | 0.991 |
| | | EPIG | 0.135 | 0.124 | 0.076 | 0.982 | 0.986 | 0.990 |
| | | TypiClust | 0.121 | 0.108 | 0.098 | 0.972 | 0.975 | 0.975 |
| | Random forest | BALD | 0.157 | 0.159 | 0.153 | 0.981 | 0.985 | 0.989 |
| | | EPIG | 0.100 | 0.093 | 0.085 | 0.981 | 0.985 | 0.989 |
| | | Random | 0.132 | 0.108 | 0.082 | 0.976 | 0.976 | 0.977 |
| | | TypiClust | 0.183 | 0.151 | 0.114 | 0.976 | 0.976 | 0.977 |
| Onehot | Linear model | Entropy | 0.152 | 0.129 | 0.107 | 0.980 | 0.981 | 0.981 |
| | | Random | 0.106 | 0.096 | 0.079 | 0.974 | 0.974 | 0.976 |
| | | TypiClust | 0.103 | 0.093 | 0.073 | 0.973 | 0.973 | 0.976 |
| | Neural network | BALD | 0.100 | 0.073 | **0.041** | 0.982 | 0.987 | 0.991 |
| | | EPIG | 0.129 | 0.108 | 0.065 | 0.982 | 0.985 | 0.990 |
| | | TypiClust | 0.134 | 0.113 | 0.084 | 0.975 | 0.976 | 0.978 |
| | Random forest | BALD | 0.162 | 0.153 | 0.139 | 0.980 | 0.984 | 0.990 |
| | | EPIG | 0.101 | 0.090 | 0.079 | 0.977 | 0.981 | 0.987 |
| | | Random | 0.207 | 0.211 | 0.142 | 0.976 | 0.976 | 0.976 |
| | | TypiClust | 0.208 | 0.210 | 0.138 | 0.976 | 0.976 | 0.976 |

**Table 1** Our experiments focus on a particular model (ESM3 encoder with random-forest prediction head) and information-theoretic data acquisition (BALD and EPIG), but we report extra results here and elsewhere in Section 6 for additional context. Here we show test metrics for `ALPS-Core-GB1` averaged over acquisition steps, across 250-2000 labels in three step ranges. Bold indicates best performance for a particular step range, underlined are not statistically significant compared to the best (by one-sided Welch's t-test).

**Data acquisition**  We focus our investigations on information-theoretic approaches to data acquisition, namely BALD (Houlsby et al, 2011) and EPIG (Bickford Smith et al, 2023), given their principled foundation and their success in recent work (Bickford Smith et al, 2023; 2024; Hübotter et al, 2024; 2025; Melo et al, 2024; Osband et al, 2023), but also because we believe they highlight a number of interesting behaviours. To provide additional context to our results, we also include TypiClust (Hacohen et al, 2022), a coverage-based method, along with uniform-random acquisition.

**Test metrics**  We measure performance using classification accuracy (higher is better) and expected negative log likelihood (NLL; lower is better) on labelled test sets. These metrics correspond to estimators of frequentist risk (Section 2) under different losses: one minus accuracy corresponds to a zero-one loss on point predictions; NLL corresponds to a log loss on probabilistic predictions.

**Active-learning loop**  We initialise the training dataset by uniform-randomly sampling two examples from each class, then we run the loop described in Section 2 until the training-label budget is used up. We run this whole process with each acquisition method at least five times with different random seeds. We report test metrics (mean ± standard error) as a function of the size of the training dataset.

### 6.2 Uncurated data helps stress-test data acquisition

We start by focusing on the `ALPS-Core` problems, in which we want to learn about the whole space of experimentally testable proteins and we can acquire a label for any protein in that space. While these represent uncurated datasets, they still arguably represent less challenging problems than some of those we consider later, as we do not need the active learning scheme to perform task adaptation, with the pool already representing the target distribution for our evaluations.

Our results show BALD and EPIG notably differing across all five problems (Figure 2). Unlike in past work (Bickford Smith et al, 2023; 2024), here we see BALD often outperforming EPIG in terms of accuracy. Interestingly, though, this does not translate into lower NLL, where BALD performs poorly and far worse than random. This likely represents a calibration issue: given the high accuracies often achieved, it seems likely that BALD is often overconfident in some of its incorrect predictions. The benefits of EPIG compared to random are also diminished when considering NLL instead of accuracy, but not to the same catastrophic degree as BALD. The root cause of these calibration issues is not immediately clear, but provides an interesting avenue for future investigation. One hypothesis is that

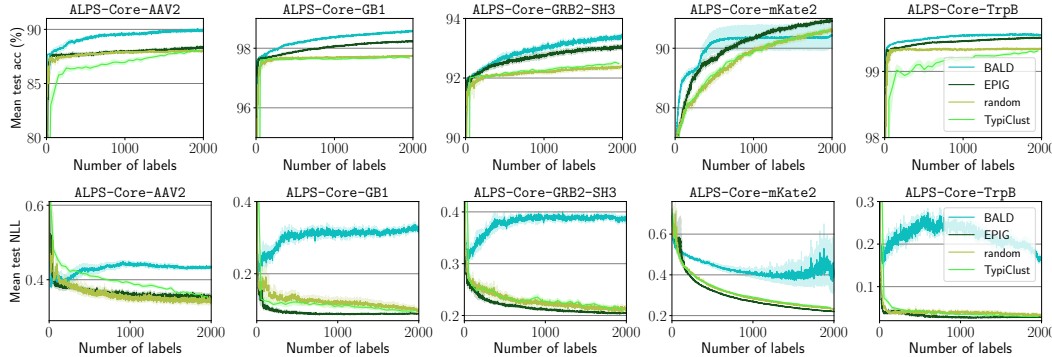

**Figure 2** Across the five `ALPS-Core` problems, the performance of information-theoretic data acquisition differs substantially depending on the quantity being targeted. BALD, which targets information in the parameters of the model being trained, consistently performs well in terms of classification accuracy but poorly as measured by negative log likelihood (NLL). EPIG, which focuses on the model's predictions, does not achieve such high accuracy in most cases but is stronger in terms of NLL, although not always better than random acquisition.

it could be connected to the statistical bias that active learning introduces (Farquhar et al, 2021), with the class ratio of actively acquired datapoints unlikely to match the original dataset.

A broader point is that success under one test metric need not translate to success under another: here we see NLL would never lead us to favour BALD but accuracy would in most cases. This highlights the need for care in choosing test metrics: neither accuracy nor NLL is "true" in an absolute sense; they simply correspond to different underlying loss functions (Section 6.1).

## 6.3 SENSITIVITY TO CLASS IMBALANCE VARIES BETWEEN ACQUISITION METHODS

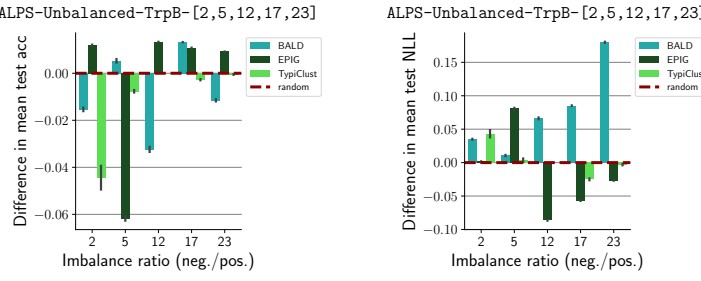

**Figure 3** The `ALPS-Unbalanced` problems demonstrate the effect of class imbalance on the performance of BALD and EPIG acquisition. BALD performs worse than random in many cases, while EPIG fails in one case.

Next we turn to the `ALPS-Unbalanced` problems. Figure 3 shows the differences in performance of EPIG and BALD relative to random as a function of the degree of class imbalance (ratio of class 0 to class 1). While there is not much of a clear trend in behaviour in terms of accuracy, we see a significant drop off in BALD's NLL performance for large imbalance ratios, whereas EPIG tends to outperform random at large levels of imbalance. As in Section 6.2, this thus highlights potential failure cases that have been underrepresented in previous active learning evaluations.

## 6.4 ACCOUNTING FOR THE TASK OF INTEREST IS KEY FOR HANDLING REDUNDANT INPUTS

Our next focus is the `ALPS-Redundant-His3` problem, in which the unlabelled pool contains a large number of redundant inputs that do not relate to the task of interest. The relative performance of EPIG on this problem shows the value of an acquisition method that explicitly accounts for the predictive task that we want to apply our model to (Figure 4). Even after acquiring 2,000 labels, BALD and random acquisition, which do not use any notion of the input distribution we wish to make predictions for, fail to reach the level of predictive performance that EPIG achieves after acquiring a handful of labels. This example thus highlights the need for active learning, and careful selection of acquisition strategies, in cases where significant task adaptation is required.

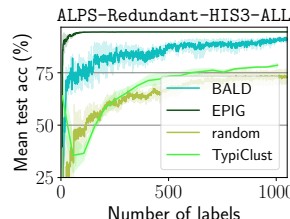
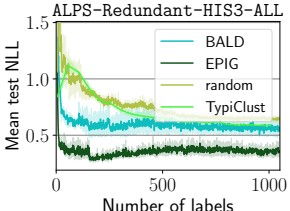

**Figure 4** The `ALPS-Redundant-His3` problem demonstrates the need for data acquisition to be targeted towards the task that we care about. EPIG acquisition, which is targeted in this way, enables fast convergence to strong performance that BALD and random acquisition cannot match even after many label acquisitions.

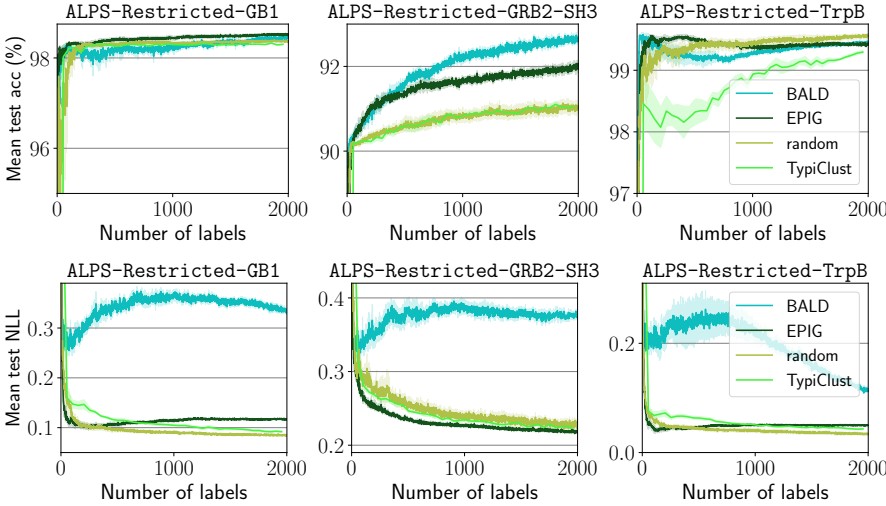

**Figure 5** EPIG can successfully gather data relevant for predictions on inputs that cannot be labelled (`ALPS-Restricted-GRB2`), but it can also fail (`ALPS-Restricted-GB1` and `ALPS-Restricted-TrpB`), underperforming random acquisition. Meanwhile BALD, lacking a notion of target inputs, tends to fail.

### 6.5 RESTRICTED ACQUISITION POSES A PARTICULARLY DIFFICULT CHALLENGE

Now we explore the particularly challenging `ALPS-Restricted` problems, which test the ability of active-learning methods to gather data to support predictions on inputs for which we cannot acquire labels directly due to limitations in the available labelling mechanism (e.g. some proteins might not be synthesisable within a given lab). Figure 5 again shows catastrophic failures for BALD in terms of NLL on all problems, with it now also failing to outperform random in terms of accuracy on two of the three problems as well. Results for EPIG appear to be quite mixed, beating random in terms of accuracy on two problems, but only in one case for NLL. It also shows an interesting behaviour in NLL where the initial performance appears strong in all cases, but then later degrades in two of them, with the NLL actually rising with increasing numbers of labels, albeit not to the extent seen by BALD. This poor performance of EPIG is perhaps surprising—as EPIG is in theory set up to accommodate these kind of transductive problems—and provides another interesting avenue for investigation.

### 6.6 STOCHASTIC BATCH ACQUISITION HAS MIXED EFFECTS ON PREDICTIVE PERFORMANCE

Next we investigate the effect of switching from acquiring one label at a time to acquiring batches of labels. For this we return to `ALPS-Core-GB1` and `ALPS-Core-TrpB`, and consider batch sizes of 16, 50 and 100. Batch acquisition is often dictated by the labelling mechanism at hand and can pose methodological challenges (Kirsch et al, 2019), but the results in Figure 6 also show it can sometimes have a protective effect against deficiencies in the acquisition function we use. In Figure 2 we saw that BALD underperformed random acquisition in terms of NLL, and here we see that its shortfall reduces at increasing batch sizes. This can be understood as a consequence of the stochastic-acquisition

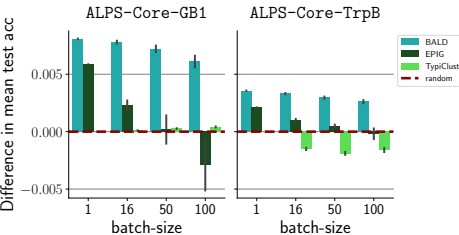 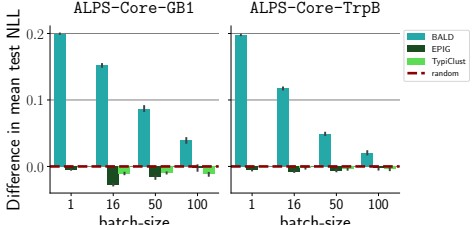

**Figure 6** Acquiring data in bigger batches leads to worse accuracy for both BALD and EPIG, but also leads to better NLL for BALD, suggesting improved calibration of the model's predictive confidence.

scheme used (Kirsch et al, 2023): as the batch size increases, we get closer to acquiring uniformly at random, which performs better than BALD-based single-label acquisition in these problems. Notably this behaviour can also have an adverse effect on performance in cases where the acquisition function is providing a good signal of data utility; the EPIG results in Figure 6 provide some evidence for this.

## 6.7 THE EFFECTIVENESS OF ACTIVE LEARNING DEPENDS ON THE MODEL BEING TRAINED

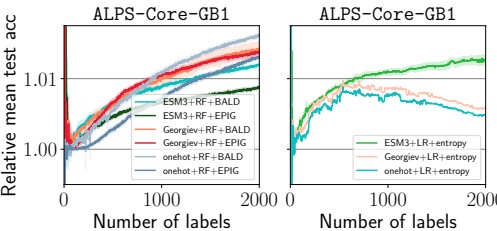 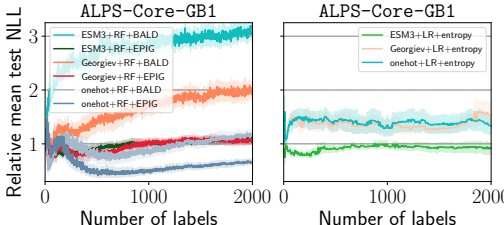

**Figure 7** The performance of an acquisition function (here we divide the test metric by the value achieved by random acquisition) can vary significantly between model configurations.

Finally we shift our focus to the configuration of the model being trained through active learning. So far we have used a combination of an ESM3 deep encoder and a random forest prediction head. Recent work suggests that this should be a good default configuration (Bickford Smith et al, 2024), but also that models based on simpler biophysical (Georgiev, 2009) and onehot encoders can perform competitively (Yang et al, 2025). We therefore return to `ALPS-Core-GB1` to investigate these alternative encoders, and also consider using a logistic-regression model instead of a random forest for the prediction head (this linear model is not stochastic and cannot be used with BALD and EPIG, so we use predictive entropy for that model instead). We find that a given acquisition function can perform well for one model and poorly for another (Figure 7). This underlines the need for evaluations to be conducted with the models that would be used in practical applications: we cannot assume results for one model class will transfer to another.

## 7 CONCLUSION

We have argued that deficiencies in existing active-learning evaluations, particularly the use of curated datasets as a starting point, can lead to a misrepresentation of methods' performance. To help address this, we have introduced `ALPS`, a set of new active-learning problems based on protein-property prediction and designed to pose challenges we believe are important for real-world deployment of active learning yet underrepresented in previous benchmarking. Our evaluations of some popular active-learning methods on `ALPS` have already raised a number of interesting new potential issues that future work might look to address, such as miscalibration of predictive uncertainty, sensitivity to class imbalance and unreliable scaling with increasing acquisition batch size. We hope that `ALPS` will not only provide a useful new testbed for active-learning researchers, but also inspire both more careful consideration of real-world issues around how methods are evaluated—and provide a stepping stone towards greater uptake of active learning in biochemistry.

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

## A  DATASETS USED IN EXISTING ACTIVE-LEARNING EVALUATIONS

From all papers published at AISTATS, ICML, NeurIPS and UAI in the past 10 years, we selected all papers with the word "active" in the title or abstract, giving an initial list of 441 papers. We then stepped through a shuffled version of this paper list, annotating each paper according to six queries:

Q1. Is the paper on active learning, based on its abstract and keywords?

Q2. Is the paper on reinforcement learning, based on its title, abstract and keywords?

Q3. What computer-vision datasets, if any, are used in the empirical evaluation?

Q4. What natural-language-processing datasets, if any, are used in the empirical evaluation?

Q5. What synthetic datasets, if any, are used in the empirical evaluation?

Q6. What other datasets, if any, are used in the empirical evaluation?

Our stopping limit was either 200 papers or four hours of annotation time; we hit the latter first, covering 103 papers within the time. We included all papers for which the answer to Q1 was "yes" and the answer to Q2 was "no".

# B ADDITIONAL RELATED WORK

**Active learning evaluations**    Active-learning methods are often assessed with established machine-learning datasets that include modifications to their composition to highlight specific methodological contributions, such as adding redundancies or class imbalances, or joining datasets (Bickford Smith et al, 2023; Citovsky et al, 2021; Lüth et al, 2023). Benchmarks developed specifically for assessing active-learning methods or assessing machine-learning models using active learning include ActiveGLAE (natural-language tasks for transformers; Rauch et al, 2023), CDALBench (combining text, vision, and tabular data; Werner et al, 2024), computer-vision tasks (Ji et al, 2023), BenchPress (code generation; Tsimpourlas et al, 2022) and Realistic-AL (Lüth et al, 2023). The last of these is arguably the closest to our work: the authors identify five "pitfalls" when applying active learning in the real world and compare their work against a number of existing efforts (Beck et al, 2021; Bengar et al, 2021; Chan et al, 2021; Gao et al, 2020; Kim et al, 2021; Krishnan et al, 2021a; Mittal et al, 2019; Munjal et al, 2022; Yi et al, 2022; Zhan et al, 2022). However, whereas Realistic-AL focuses its analysis on large, labelled computer-vision datasets (eg, CIFAR-10 and MIO-TCD)—as do the studies they compare against—our ALPS problems shift the focus to the domain of protein-property prediction. In this focus on a scientific application, our work aligns with that of Gorantla et al (2024), who applied several active-learning methods to binding-affinity-prediction tasks.

**Protein-property prediction**    Predicting the structure and the function of a protein from its sequence is an important challenge in biochemistry, and its principled assessment (e.g., CASP; Kryshtafovych et al, 2019) has facilitated significant progress by machine learning in the field (AlQuraishi, 2019; Jumper et al, 2021; Evans et al, 2021; Abramson et al, 2024). Over recent years, benchmarking property prediction has seen many efforts curating publicly available datasets and tasks (Dallago et al, 2021; Frazer et al, 2021; Kucera et al, 2024; Notin et al, 2023; Rao et al, 2019; Riesselman et al, 2018; Xu et al, 2022). Notable property-prediction benchmarks include FLIP (Dallago et al, 2021), PEER (Xu et al, 2022), ProteinGym (Notin et al, 2023) and ProteinShake (Kucera et al, 2024). These efforts have generally been tailored to assess (static) machine-learning models' predictive performance (zero-shot predictions of mutation effects in clinical and deep-mutational-scanning assays). Thus, none of the previous protein-property benchmarks have assessed active-learning methods for their usability or elucidated algorithmic properties and shortfalls. The use of probabilistic models and Bayesian-optimisation algorithms to optimise one or multiple protein properties has been considered in (Romero et al, 2013; Stanton et al, 2022; Gruver et al, 2023; Khan et al, 2023). Finally, Yang et al (2025) optimised protein properties using active learning (effectively performing batch Bayesian optimisation) but did not focus on evaluation design in its own right.

| Identifier | Validation-set cost | Hyperparameter | Modality | Acquisition | References |
|---|---|---|---|---|---|
| Realistic-AL | ✓ | ✓ | image | batch | Lüth et al (2023) |
| ActiveGLAE | ✓ | ✓ | text | batch | Rauch et al (2023) |
| LabelBench | ✓ | ✓ | image | batch | Zhang et al (2024) |
| CDALBench | ✓ | ✓ | text, image, tabular | single,batch | Werner et al (2024) |
| Reliable deep AL | ✗ | ✓ | image | batch | Ji et al (2023) |
| BenchPress* | ✗ | ✗ | code generation | | Tsimpourlas et al (2022) |
| DISTIL | ✗ | ✗ | image | batch | Beck et al (2021) |
| Reducing label effort | ✗ | ✗ | image | batch | Bengar et al (2021) |
| Marginal benefit of AL | ✗ | ✗ | image | | Chan et al (2021) |
| Consistency-based semi-supervised AL | ✗ | ✗ | image | batch | Gao et al (2020) |
| TA-VAAL | ✗ | (✓) | image | batch | Kim et al (2021) |
| SCAL | ✗ | ✗ | image | batch | Krishnan et al (2021b) |
| Parting with illusions | ✗ | ✗ | image | batch | Mittal et al (2019) |
| Robust & reproducible AL | ✓ | ✓ | image | batch | Munjal et al (2022) |
| PT4AL | ✗ | ✗ | image | single,batch | Yi et al (2022) |
| DeepAL+ | ✗ | ✗ | image | batch | Zhan et al (2022) |
| Revisiting AL, Vision Foundation | No* | No* | image | batch | Gupte et al (2024) |
| BADGE | ✗ | ✓ | image, openml | batch | Ash et al (2020) |
| Interplay of UM and deep AL | ✓ | ✓ | image, synth | batch | Huseljic et al (2024) |
| Open-Set annotation, LfOSA | ✗ | ✗ | image | batch | Ning et al (2022) |
| AL for imbalanced datasets | Yes* | ✓ | image | batch | Aggarwal et al (2020) |
| Limitations of AL | ✗ | ✗ | image, text | batch | Hu et al (2021) |
| optimal AL | ✗ | ✗ | image, text | batch | Zhou et al (2021) |
| Benchmarking pool-based AL | ✗ | ✗ | tabular, synth | single, batch | Zhan et al (2021) |
| Efficacy of deep AL for image | ✗ | ✗ | image | batch | Li et al (2022) |
| Margin all you need? | ✗ | ✗ | tabular | batch | Bahri et al (2022) |
| Ours | ✓ | ✓ | biochemistry | single,batch | |

**Table 2** Comparison of related Active Learning benchmarks with emphasis on the inclusion of validation set cost, hyperparameter-tuning, and covered modalities. Noteable exceptions: BenchPress is a code generation framework and not strictly an AL benchmark. Special cases (marked with *) include (Gupte et al, 2024), which acknowledge the issues but do not discuss solutions in their benchmark, and (Aggarwal et al, 2020), which considers 10-fold cross-validation.

# C ALPS DETAILS

## C.1 SOURCE DATA

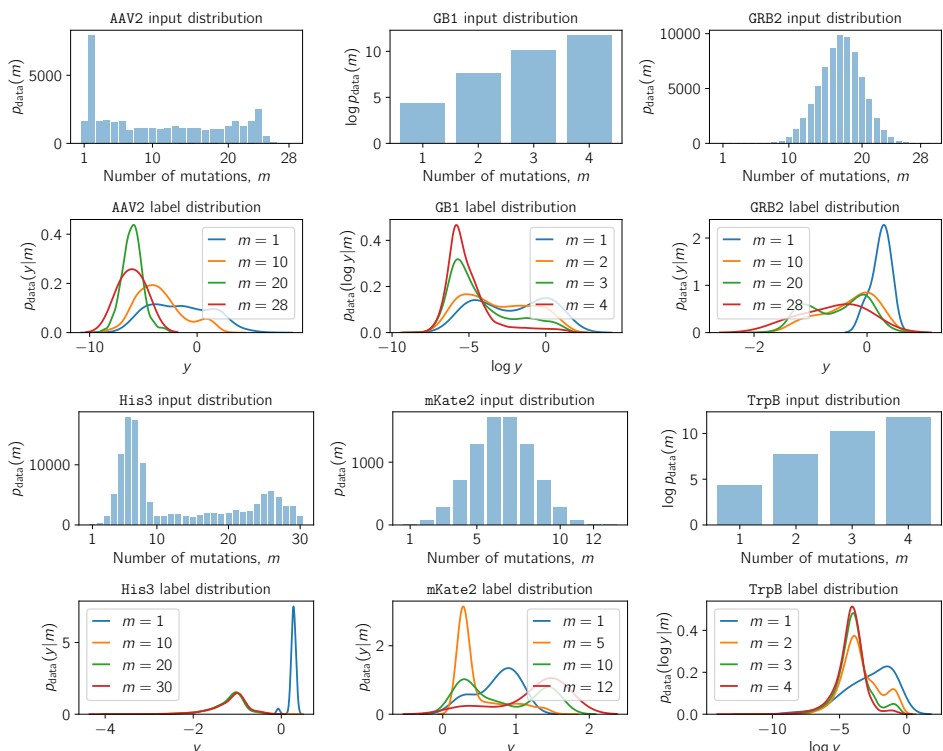

**Figure 8** Overview of the label and property distributions for all source datasets that are the basis for the `ALPS` problems. The number of mutations (as properties of the sequence) was used to curate the `ALPS-Restricted` tasks, whereas the labels have been used to curate class balances for `ALPS-Unbalanced`. See Table 3 for references and measured effects.

| Dataset | Effect of Interest | Description (number sites mutated) (unique positions) | N | References | License |
|---|---|---|---|---|---|
| GB1 | Epistasis (indiv.) | combinatorially complete binding (0-4) (4) | 149,361 | Wu et al (2016); Yang et al (2025) | CC-BY 4.0 International |
| TrpB | Epistasis (indiv.) | combinatorially complete enzyme (0-4) (4) | 159,129 | Johnston et al (2024); Yang et al (2025) | CC-BY 4.0 International |
| GRB2-SH3 | Allostery (design) | allosteric abundance+binding library, (0-20) (34) | 71,233 | Faure et al (2024) | MIT |
| AAV2 | Viability (design) | engineered viral capsid (0-29) (varying lengths) | 39,172 | Dallago et al (2021); Bryant et al (2021) | MIT |
| mKate2 | Epistasis (general) | bridging two genotypes (eqFP) (0-13) | 8,192 | Poelwijk et al (2019); Faure & Lehner (2024) | CC-BY 4.0 International |
| His3 | Epistasis (general) | 12 WTs with high-order mutants (NA) | 956,648 | Pokusaeva et al (2019); Notin et al (2023) | CC-BY 4.0 International |

**Table 3** ALPS source data overview, displaying investigated effect (from the original source), number of samples in the data, and license. For `AAV2`, a subset of random mutagenesis deselecting model-dependent designs was used.

## C.2 PROBLEMS

Generally, we define a task in the `ALPS` benchmark based on the label set, or any property vector which can be derived from the input sequence, see Figure 8. We specifically consider the Hamming distance relative to the reference (wild-type) sequence. To further specify the task, we can curate the label vector and the property vector by sub-selecting either or both. This allows us to easily add new tasks if required, based on the label-set or input properties.

### C.2.1 CORE TASK

A broad test bed for the label acquisition strategies is the datasets in their raw, uncurated form. These datasets allow us to test the hypothesis whether active learning applies in uncurated, imbalanced experiment settings. We consider 20% of all samples distinct from pool/training for testing, to reflect

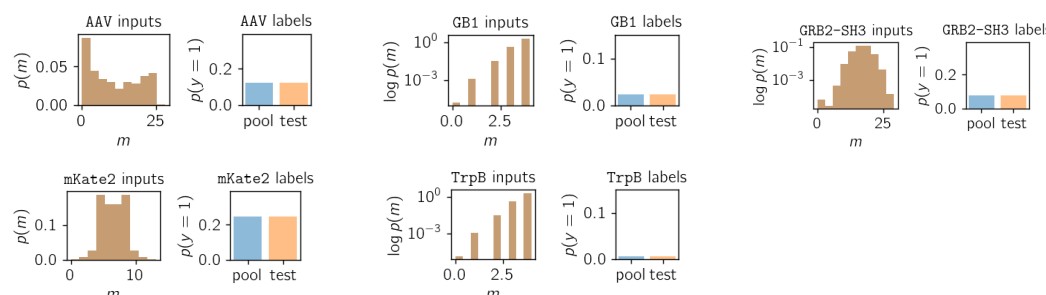

(a) `Core` task pool, test label-sets. Both label and input property distributions are the same between pool and test sets.

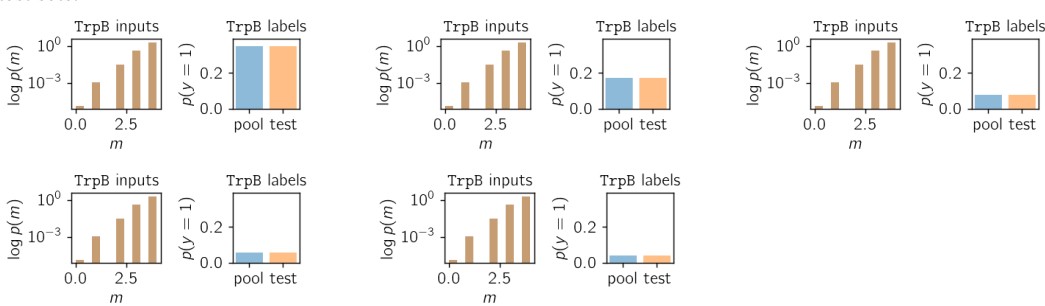

(b) `Unbalanced` task pool, test label-sets. The label and input property distributions are the same between pool and test sets.

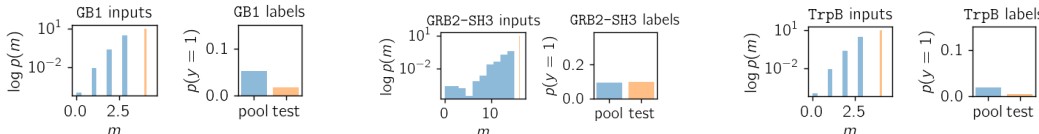

(c) `Restricted` task pool, test label-sets. Both label and input property distributions change between pool and test set.

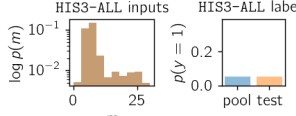

(d) `Redundant` task pool, test label-sets. Input and label distributions are the same for pool and test set.

**Figure 9** Overview of the label distributions. For each task and source data the input property mutation $m$ count as density (left) and likelihood of observing the positive label (right). Mutations for `His3` have been taken from the source set with respect to one reference sequence ("Scer").

standard AL setups. We also consider a fully uncurated setup, the pool set consists of all input-label pairs available, and the task is to predict the labels on the complete dataset Appendix D.1. For this specific case, $\mathcal{X}_{\text{pool}} = \mathcal{X}_{\text{test}}$ and any input, label pair, which we acquire from the pool is present in the test set; translating into $\mathcal{X}_{\text{train}} \subset \mathcal{X}_{\text{test}}$. This task (in either configuration) does not apply to `His3`, since binarization requires considerable post-processing steps, which can be found in the original reference Pokusaeva et al (2019).

### C.2.2 UNBALANCED TASK

We investigate whether a change in threshold affects algorithm performance, apart from the reference inputs used to measure improvement against, initially motivated by protein engineering practices. Our task follows the previously described uncurated, imbalanced (`Core`) setup where $\mathcal{X}_{\text{train}} \subseteq \mathcal{X}_{\text{pool}} = \mathcal{X}_{\text{test}}$. To obtain varying degrees of label imbalance, we compute five constant threshold values equidistant between the median label and previous (WT) reference values Figure 9b. We do so by

| Source Task | transform | AAV2 | GB1 | GRB2-SH3 | mKate2 | His3 | TrpB |
|---|---|---|---|---|---|---|---|
| uncurate | binarize (ref.) | Core | Core | Core | Core | see Pokusaeva et al (2019) | Core |
| imbalance | binarize (const.) | Unbalanced | Unbalanced | Unbalanced | Unbalanced | – | Unbalanced |
| constraint/transfer | subselect property | Restricted | Restricted | Restricted | Restricted | – | Restricted |
| redundancies | binarize & assign UNK | – | – | – | – | Redundant | – |

**Table 4** Overview of the problems and what task they are addressing (index), what transformation is applied to the source labels (columns) to obtain the problem (cells). We report results for individual problems (black) and if task definitions apply (in general) and can be derived with the provided code-base they are indicated in gray. For example, it is possible to define `Restricted` task(s) for any input sequence if there are more than two distinct set of mutations with enough samples to account for the acquisition budget. However, not all distinct sets of mutations with which a restricted task can be defined present plausible (practical) scenarios. Across all tasks `His3` presents an exception, as the unprocessed measurements cannot be used for the `Core` task, and due to the nature of the source data has to be treated with care Pokusaeva et al (2019), see Appendix C.2.4.

discretizing with constant values, for `TrpB` specifically $t \in [-4, -3.5, -3, -2.75, -2.5]$ in $\log y$ labels (see `alps/config/compute_protein_task/data/trpb.yaml`).

```
- labels: binary_const
  const_val: -4
```

### C.2.3 RESTRICTED TASK

We split the dataset into a disjoint pool- and test-set, to test whether active label acquisition is beneficial when an out-of-domain test/target set is given. The objective is to predict labels from inputs with $k$ number of mutations relative to the reference. Given Hamming distance $HD$ of the string inputs $x$, let $\mathcal{X}_{\text{test}} = \{x \in \mathcal{X} \mid HD(x, x_{\text{ref}}) = k\}$ and $\mathcal{X}_{\text{pool}} = \{x \in \mathcal{X} \mid HD(x, x_{\text{ref}}) < k\}$. The pool from which training labels are acquired has $< k$ mutations. The test set with which we assess performance has $k$ mutations. This subsequently yields different label distributions between pool and test Figure 9c. To obtain the run configurations via the described experimental specifications, we set the `GB1`, `TrpB` tasks like so

```
- labels: binary_wt
    curated: True
    subset_by: k_mutations
    subset_classes:
      - [0, 1, 2, 3]
      - [4]
```

This task reflects an experimental measurement campaign, where over multiple rounds more mutations are introduced to the inputs and a proposal model is used to predict the next set of variants, describing a transfer learning setting of the predictive models. Alternatively, this task can be formulated by selecting other label or inputs sets, for example discretizing labels into multiple quantiles and assigning pool and test to different quantile classes.

#### C.2.4 REDUNDANT TASK

The task we specify encompasses a $\mathcal{X}_{\text{pool}}$ of largely uninformative labels and a labelled (zero-one) minority set. The setting under which labels have been obtained (`His3`) reflects redundancy in the pool due to multiple references when measuring observations. Specifically, labels have been obtained for different experimental setups (libraries). Therefore, discretizing with respect to one reference becomes impossible across all data in the source set. Given that inputs in `His3` are associated with multiple wild-types, measured under different experimental conditions, we binarize one library with one reference input (the one it has been compiled with) and assign a third class to all other observations (the remaining 11 libraries). We refer to this third class as "neither" in the manuscript. Our pool consists of $\approx 86.8\%$ uninformative (third class) samples, and the labeled classes are $\approx 12.4\%$ negative and $\approx 0.7\%$ positive labels. The starting training pool contains two labels for each of the three classes. The pool contains all three classes, while test contains two classes only. To replicate the experiment setup within the benchmark suite requires to first discretize with three classes, and then to subselect the test classes of interest.

See `alps/config/compute_protein_tasks/data/his3.yaml` which targets `alps/src/data/tasks/pg.py`. To obtain the run configurations via the experimental specifications, we use

```
- data_item:
    id: HIS3-ALL
    wildtype_sequence: EALGAVRGVKRFGSGFAPLDEALSRAVVDL
    positions: []
    data_dir: ${directories.data}/his/S_all_scaled_info_v2.csv
  curated: True
  subset_by: labels
  labels: ternary_wt
  subset_classes:
    - [0, 1, 2]
    - [0, 1]
  target_id: HIS3-S1
```

#### C.2.5 BATCHED TASK

The underlying tasks are the `Core` (uncurated) setups, see Appendix C.2.1, however the acquisition algorithms are run with `batch_size`>1 (as specified).

### C.3 ENCODINGS

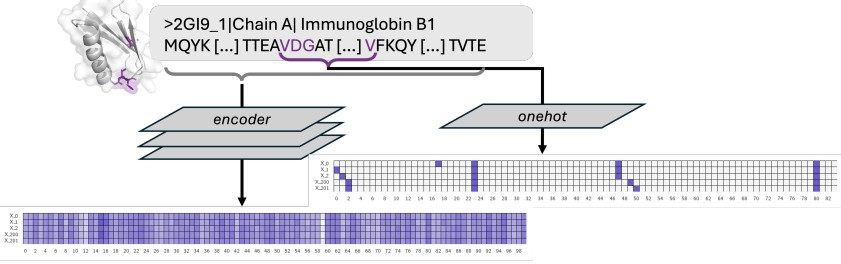

**Figure 10** Encoding of the protein sequence inputs. We encode a reference sequence, consisting of single-letter amino acid codes, into a continuous vector of fixed length. Pretrained encoder models take the full sequence length (for all datasets) and encode it to the model's number of dimensions, see Table 5. The `onehot` and `georgiev` encodings can be limited to encode only the mutated positions (`GB1`,`TrpB`), segments including mutations (`mKate2`,`His3`), and full-length sequences (`GRB2-SH3`,`AAV2`).

The inputs are string sequences (amino acid sequences) either as the residues of the mutated positions (`GB1`,`TrpB`) or the sequence of complete length (`GRB2`,`AAV2`,`His3`), which we encode to a real-valued matrix. To select the best performing protein language model (PLM) we evaluate a set of 22

PLMs available through huggingface, Table 5. As simple baselines, we consider simple onehot and georgiev encodings of the amino acid sequences Georgiev (2009).

| Name | #dimensions | #layers | #params | Memory | Dataset | Reference |
|---|---|---|---|---|---|---|
| esm1_t6_43M_UR50S | 768 | 6 | 43M | 0.17GB | Uniref50/S 2018_0 | Rives et al (2021) |
| esm1_t12_85M_UR50S | 768 | 12 | 85M | 0.34GB | Uniref50/S 2018_0 | Rives et al (2021) |
| esm1_t34_670M_UR50D | 1280 | 34 | 670M | 2.7GB | Uniref50/D 2018_0 | Rives et al (2021) |
| esm1_t34_670M_UR50S | 1280 | 34 | 670M | 2.7GB | Uniref50/S 2018_0 | Rives et al (2021) |
| esm1_t34_670M_UR100 | 1280 | 34 | 670M | 2.7GB | Uniref100 2018_0 | Rives et al (2021) |
| esm1b_t33_650M_UR50S | 1280 | 33 | 650M | 2.6GB | Uniref50/S 2018_0 | Rives et al (2021) |
| esm1v_t33_650M_UR90S_[1-5] | 1280 | 33 | 650M | 2.6GB | Uniref90/S 2020_0 | Meier et al (2021) |
| esm2_t6_8M_UR50D | 320 | 6 | 8M | 0.03GB | Uniref50/D 2021_0 | Lin et al (2023) |
| esm2_t12_35M_UR50D | 480 | 12 | 35M | 0.14GB | Uniref50/D 2021_0 | Lin et al (2023) |
| esm2_t30_150M_UR50D | 640 | 30 | 150M | 0.6GB | Uniref50/D 2021_0 | Lin et al (2023) |
| esm2_t33_650M_UR50D | 1280 | 33 | 650M | 2.6GB | Uniref50/D 2021_0 | Lin et al (2023) |
| esm2_t36_3B_UR50D | 2560 | 36 | 3B | 12GB | Uniref50/D 2021_0 | Lin et al (2023) |
| esm2_t48_15B_UR50D | 5120 | 48 | 15B | 60GB | Uniref50/D 2021_0 | Lin et al (2023) |
| esm3_sm_open_v1 | 1536 | 48 | 1.4B | 5.6GB | custom | Hayes et al (2025) |
| prot_albert | 4096 | 12 | 224M | 0.9GB | Uniref100 | Elnaggar et al (2022) |
| prot_bert | 1024 | 30 | 420M | 1.7GB | Uniref100 | Elnaggar et al (2022) |
| prot_bert_bfd | 1024 | 30 | 420M | 1.7GB | BFD100 | Elnaggar et al (2022) |
| prot_xlnet | 1024 | 30 | 409M | 1.6GB | Uniref100 | Elnaggar et al (2022) |
| prot_t5_xl_uniref50 | 1024 | 24 | 3B | 12GB | Uniref50 | Elnaggar et al (2022) |
| prot_t5_xl_bfd | 1024 | 24 | 3B | 12GB | BFD100 | Elnaggar et al (2022) |
| prot_t5_xxl_uniref50 | 1024 | 24 | 11B | 44GB | Uniref50 | Elnaggar et al (2022) |
| prot_t5_xxl_bfd | 1024 | 24 | 11B | 44GB | BFD100 | Elnaggar et al (2022) |

**Table 5**  Overview of all pretrained encoders available in `ALPS`.

### C.4  LABEL PREPROCESSING

**Binary classification**  Given a reference threshold, listed *WT* reference sequence value (unless indicated otherwise), we assign positive classes if function values are equal or greater than that reference value.

Exact specifications for reference sequence `wildtype_sequence` (in sets of sequences `seq_id`) and labels (`label_id`) can be found in the respective `alps/config/compute_protein_tasks/data/` `{aav,allo,eqfp,gb1,grb2,his3,trpb}.yaml` . The *WT* reference binary classification is `labels:  binary_wt`.

Binary classes can also be assigned by **constant** values, see `labels:  binary_const`, which has been applied to compute `ALPS-Unbalanced`.

### C.5  METRICS

**Accuracy**  Given $N$ input-label pairs, $(x_*^i, y_*^i)_{i=1}^N$, we compute

$$\text{accuracy} := \frac{1}{N} \sum_{i=1}^N \mathbb{I}(\arg\max_{y_*'} p_\phi(y_*'|x_*^i) = y_*^i). \tag{1}$$

**Expected negative log likelihood**  We compute

$$\text{NLL} := -\frac{1}{N} \sum_{i=1}^N \log p_\phi(y_*' = y_*^i | x_*^i). \tag{2}$$

**F1 score**  We compute the F1 score from the true positive count (TP) and false positive count (FP) as

$$f_1 := \frac{2\text{TP}}{(2\text{TP} + \text{FP} + \text{FN})}. \tag{3}$$

**AUROC**  We use the Scikit-learn (Pedregosa et al, 2011) implementation to calculate AUROC (`macro` (unweighted) aggregate with McClish correction). For the binary-label case, we compute

$$\text{ROC-AUC} := \frac{1}{2}\left(1 + \frac{\text{AUC}(\text{FPR}, \text{TPR}) - \frac{1}{2}\max(\text{FPR})^2}{\max(\text{FPR}) - \frac{1}{2}\max(\text{FPR})^2}\right). \tag{4}$$

## C.6 Algorithms

### C.6.1 Prediction heads

**Logistic regression** is as implemented in scikit-learn (`sklearn.linear_model.LogisticRegression`) (`max_iter=10000`) with $l_2$ penalty, optimized regularization parameter `C`, given a validation sample. For each model fit, we determine the optimal regularizer $\in [0.001, 0.01, 1, 100, 1000]$ as minimizing $\mathcal{L}_{\text{NLL}}$ on the validation set.

**Random forest** is as implemented in scikit-learn (`sklearn.ensemble.RandomForestClassifier`) with default parameters (`n_estimators=100` using the `gini` criterion).

**Neural network with MC dropout** implemented in PyTorch, three layer fully connected architecture (sizes 128, 128, 128) with dropout-rate of 0.1 (10%), following Gal & Ghahramani (2016) `https://github.com/yaringal/DropoutUncertaintyExps`. Training is done minimizing $\mathcal{L}_{\text{NLL}}$ loss (unless stated otherwise) with early stopping (patience is 5.000 steps) on a validation set (size 1.000 samples). Optimizer is (PyTorch's) `SGD` optimizer with learning-rate $\gamma = 0.01$ and weight decay $\lambda = 0.0001$ Paszke et al (2019).

### C.6.2 Acquisition

**EPIG** as implemented in Bickford Smith et al (2023) (available at `https://github.com/fbickfordsmith/epig` under MIT license) with `n_target_samples=100` *without* nested MC computed from scores in batches of 1000.

**BALD** as implemented in Kirsch et al (2019) (available at `https://github.com/BlackHC/batchbald_redux/` under Apache-2.0 license) computing scores in batches of 1000.

**Random** Random acquisition is `numpy` (v1.26.0) random (Generator) `choice` (without replacement) with `size=batch_size`.

**TypiClust** follows the implementation in Hacohen et al (2022) as provided in the repository (`https://github.com/avihu111/TypiClust/`) (MIT license) with `n_neighbors=20`. A batch size of 50 is used, unless stated otherwise.

**BADGE** follows the implementation in Ash et al (2020) from the repository (`https://github.com/JordanAsh/badge/`) and is applied to neural network predictors with MC-dropout, unless indicated otherwise. Due to run-time of the underlying models a batch-size of 50 is used, unless indicated otherwise.

**BAIT** follows the implementation in Ash et al (2021) and is used with neural network prediction heads and a batch-size of 50, unless indicated otherwise.

# D ADDITIONAL RESULTS

## D.1 POOL EQUAL TO TEST

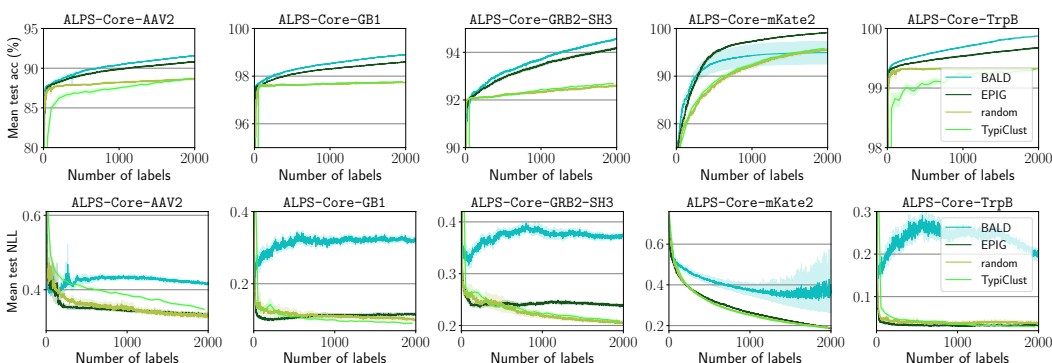

**Figure 11** `Core` task performance with test-set as all samples (including training). Generally, we find an increase in test performance for all acquisitions BALD, EPIG, and random (cf. Fig 1) and a decreased standard error across seeds (7 seeds reported). The order of performance stays the same, i.e. BALD outperforming EPIG in 4 out of 5 core tasks, and EPIG outperforming BALD on all `Core` tasks for the expected NLL loss.

## D.2 ADDITIONAL CLASSIFIER

We include a fully connected neural network (three layers) with MC dropout (rate 10%) Gal & Ghahramani (2016) as a deep learning classifier. As this significantly increases the compute time per step, we set the batch size to 50 and report results on two core datasets `GB1,TrpB`.

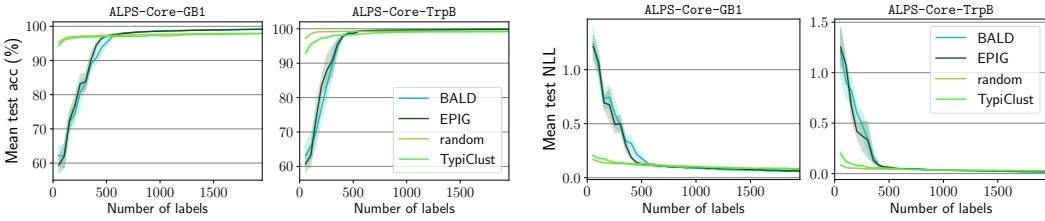

**Figure 12** Performance of the neural network (NN) model with MC dropout (ESM3 encoding) on two `Core` tasks (`GB1,TrpB`) using batch acquisition (BALD, EPIG, TypiClust) in batches of 50 (8 seeds). Compared to the random forest (RF) performance on the same datasets, we ultimately observe very high accuracy and comparable NLL values. However, with a NN, both BALD and EPIG require more labels to obtain the same performance, i.e. up to 400 labels the test accuracy is below 90, which is significantly lower than for ESM3+RF on the same pool and test set also using batch acquisition Figure 13.

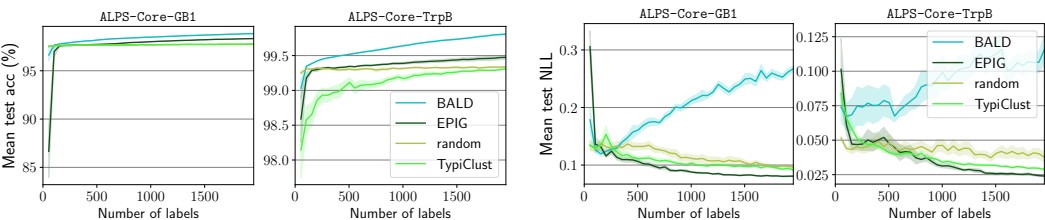

**Figure 13** Performance of random forest prediction (ESM3 encoding) using batched acquisition (BALD, EPIG) with batch-size 50 (8 seeds).

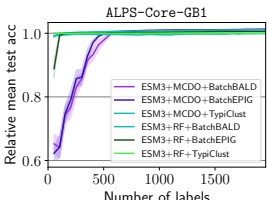
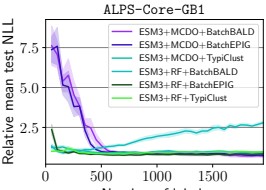

**Figure 14** Relative performance over random (ratio) comparing neural network prediction head with MC dropout against random-forest regressor on ESM3 on one `Core` task (`GB1`) using batched acquisition (BALD, EPIG, TypiClust) with batch-size 50 (8 seeds).

### D.3 F1-SCORE METRIC

Core tasks presented with the F1-score metric.

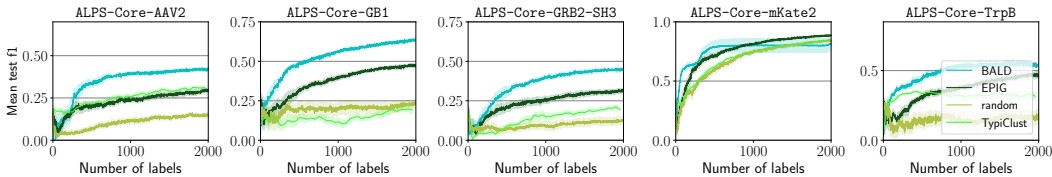

**Figure 15** Performance (F1 score on the test set) of a random-forest classifier on ESM3 over number of acquired labels (x-axis). We observe higher test performance of active acquisition (EPIG, BALD) over random (light green) except for curated set (`mKate2`).

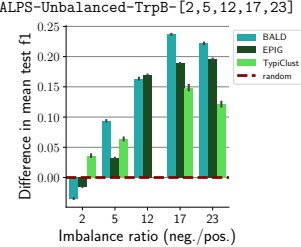

**Figure 16** Performance (mean F1 score on test computed over all acquired samples with std.err.) over different imbalance ratios (x-axis) (zero to one proportion) obtained from varying threshold discretization (six seeds).

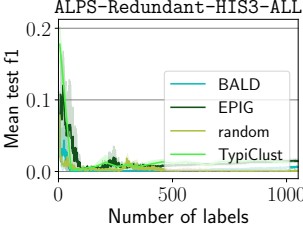

**Figure 18** Test performance (F1 score) on two-class test set, with three class pool/training set. Prediction-oriented active learning shows significant gains over random acquisition.

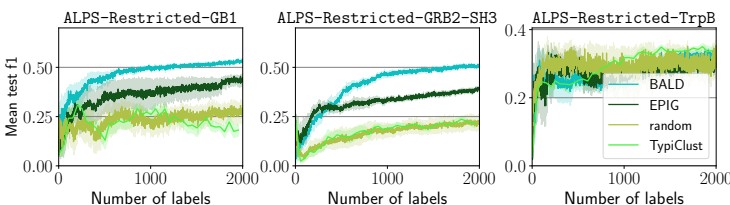

**Figure 17** Performance (test F1 score) with ESM3 encoded inputs for training on pool distinct from the test set on seven seeds. sTraining inputs are up to $HD = (k - 1)$ from a reference, and test/target set is $HD = k$, with $k = 4$ for `GB1`, `TrpB`, and $k = 16$ for `GRB2-SH3`.

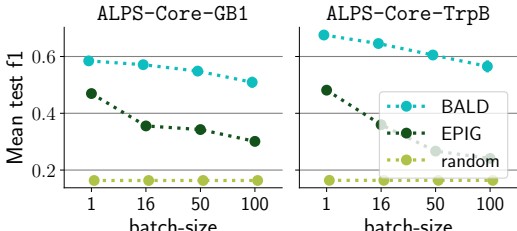

**Figure 19** Expected performance (empirical mean F1 score across steps, with standard error across 8 seeds) over different batch-sizes (x-axis).

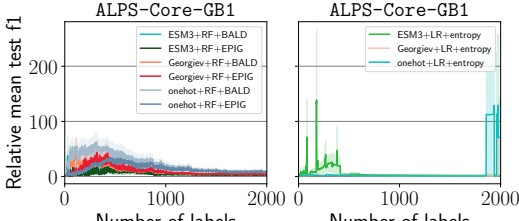

**Figure 20** Test performance (F1 score) relative to random performance (ratio) for all models (three encoders with two classifiers) over number of acquired samples.

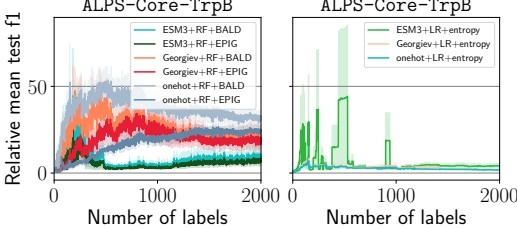

**Figure 21** Test performance (F1 score) for all models (three encodings with two classifiers) over number of acquired samples.

## D.4 AUROC METRIC

All experimental results presented with the AUROC metric.

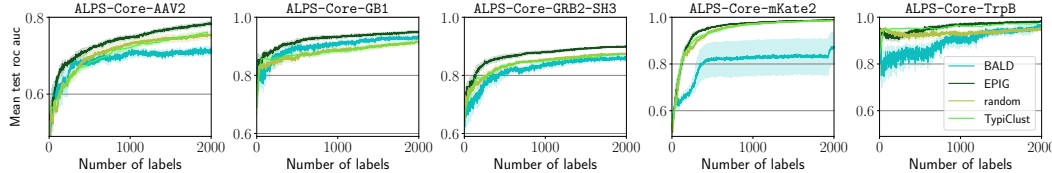

**Figure 22** Performance (AUROC) of a random-forest classifier on ESM3 on the test set over number of acquisitions. We observe higher test performance of active acquisition (EPIG) over random (light green) except for `AAV2` (8 seeds).

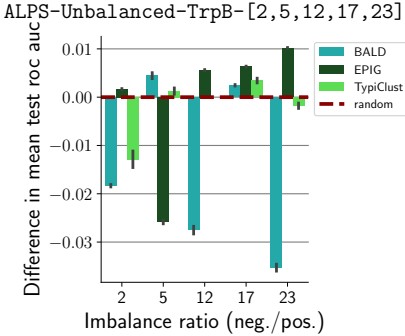

**Figure 23** Performance of test AUROC (random-forest on ESM3, mean over run with std.err.) given different imbalance-ratios (zero-to-one) by varying threshold discretization (six seeds).

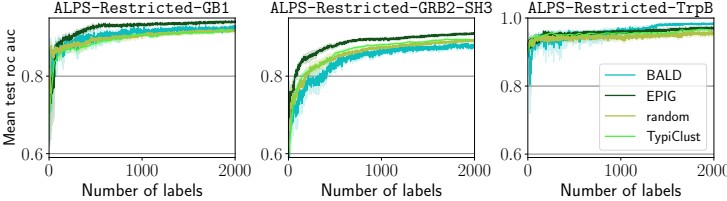

**Figure 24** Performance (AUROC) with random-forest on ESM3 encoded inputs for training on pool distinct from the test set. Training inputs are up to $HD = (k-1)$ from a reference, and test/target set is $HD = k$, with $k = 4$ for `GB1`, `TrpB`, and $k = 16$ for `GRB2-SH3` (8 seeds).

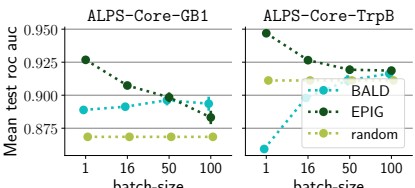

**Figure 25** Expected performance (empirical mean over test metric across steps, with standard error (8 seeds) over different batch-sizes.

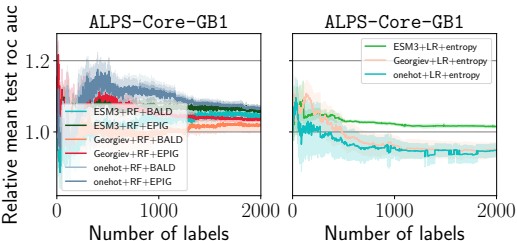

**Figure 26** Relative performance (test AUROC to random ratio) for all models (three encoders with two classifiers) over the number of acquired samples.

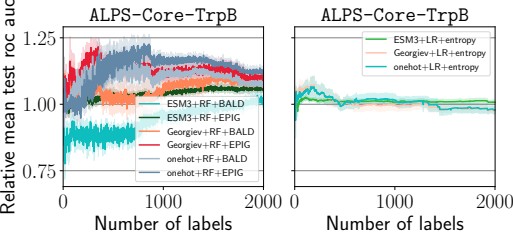

**Figure 27** Test performance (AUROC) for all models (three encoders with two classifiers) over the number of acquired samples.

## D.5 BADGE AND BAIT

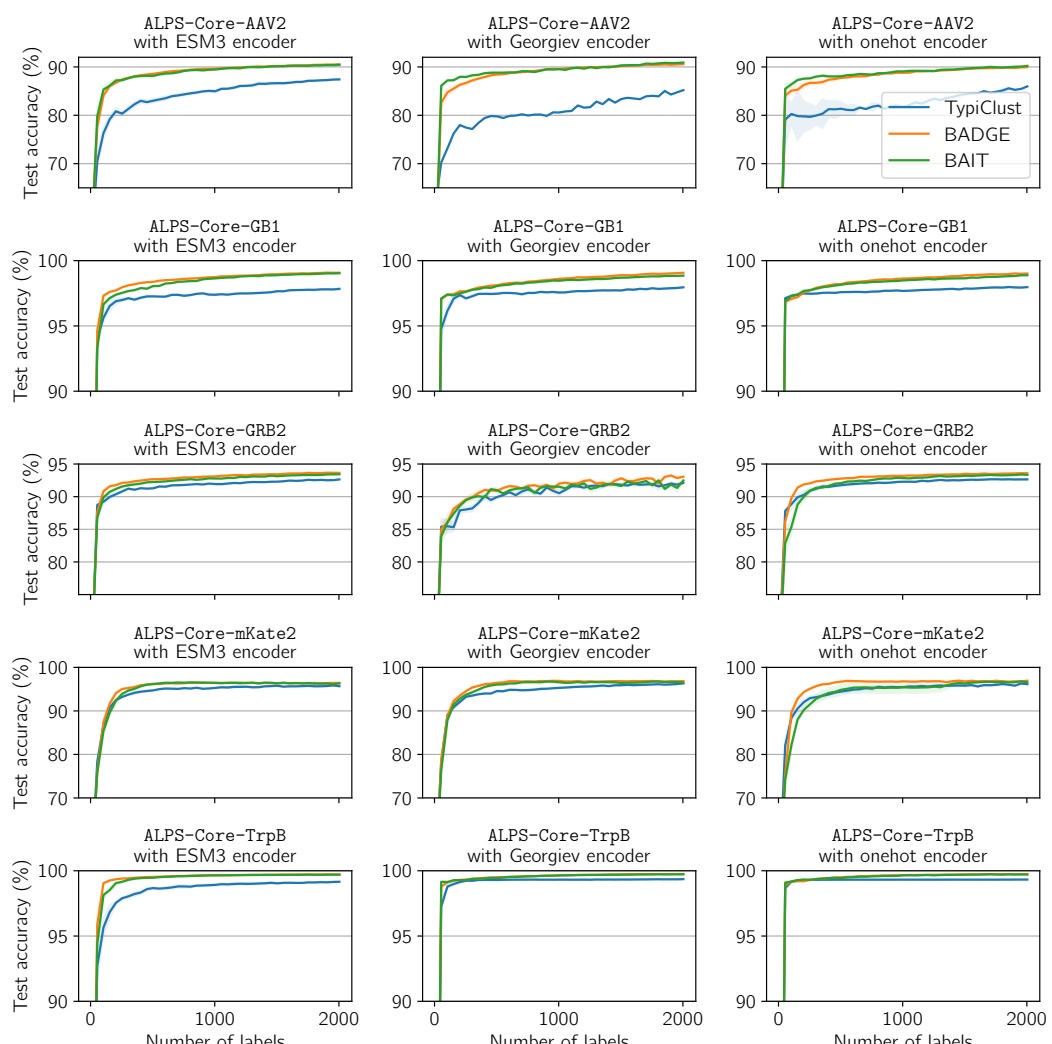

**Figure 28** Test accuracy of BADGE and BAIT relative to TypiClust on the `ALPS-Core` problems.

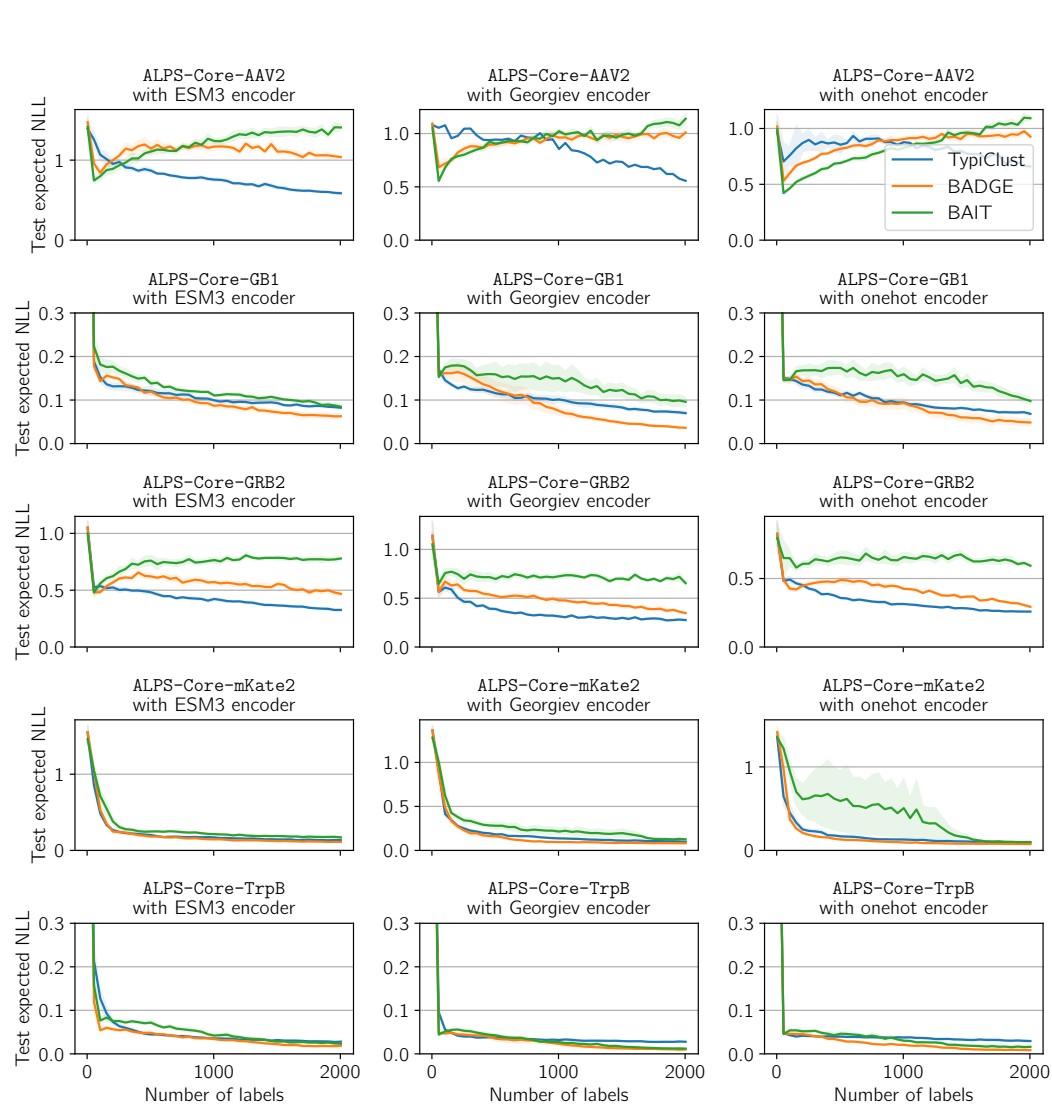

**Figure 29** Test expected NLL of BADGE and BAIT relative to TypiClust on the `ALPS-Core` problems.

## D.6 RUN TIMES

| Dataset | Prediction head | Method | Median | Minimum | Maximum |
|---|---|---|---|---|---|
| AAV | Random forest | BALD | 10 | 10 | 10 |
| | | EPIG | 8 | 6 | 12 |
| | | TypiClust | 231 | 226 | 279 |
| | | Random | 5 | 4 | 7 |
| | Neural network | BADGE | 86 | 77 | 104 |
| | | BAIT | 75 | 67 | 89 |
| | | TypiClust | 304 | 298 | 352 |
| | | Random | 70 | 62 | 94 |
| GB1 | Random forest | BALD | 21 | 15 | 32 |
| | | EPIG | 30 | 19 | 30 |
| | | TypiClust | 263 | 166 | 265 |
| | | Random | 7 | 7 | 7 |
| | Neural network | BADGE | 123 | 107 | 3519 |
| | | BAIT | 117 | 112 | 487 |
| | | TypiClust | 354 | 252 | 356 |
| | | Random | 115 | 94 | 127 |
| GRB2 | Random forest | BALD | 15 | 8 | 18 |
| | | EPIG | 15 | 10 | 19 |
| | | TypiClust | 242 | 168 | 248 |
| | | Random | 8 | 7 | 9 |
| | Neural network | BADGE | 85 | 81 | 106 |
| | | BAIT | 84 | 81 | 102 |
| | | TypiClust | 312 | 243 | 318 |
| | | Random | 87 | 74 | 95 |
| TrpB | Random forest | BALD | 16 | 15 | 28 |
| | | EPIG | 21 | 16 | 28 |
| | | TypiClust | 263 | 170 | 345 |
| | | Random | 6 | 5 | 10 |
| | Neural network | BADGE | 137 | 112 | 2954 |
| | | BAIT | 132 | 115 | 1091 |
| | | TypiClust | 361 | 316 | 467 |
| | | Random | 134 | 115 | 151 |
| mKate2 | Random forest | BALD | 5 | 5 | 6 |
| | | EPIG | 5 | 4 | 6 |
| | | TypiClust | 117 | 75 | 119 |
| | | Random | 4 | 4 | 6 |
| | Neural network | BADGE | 78 | 72 | 97 |
| | | BAIT | 65 | 62 | 80 |
| | | TypiClust | 164 | 134 | 171 |
| | | Random | 76 | 72 | 81 |

**Table 6** Per-step (acquisition plus training) run times (in seconds) on the `ALPS-Core` problems.

## D.7 PLOTS WITH FULL VERTICAL-AXIS RANGES

Some of the active-learning plots in Section 6 use vertical axes with reduced ranges so that the gaps between curves are easier to see. Here we present corresponding plots with full ranges.

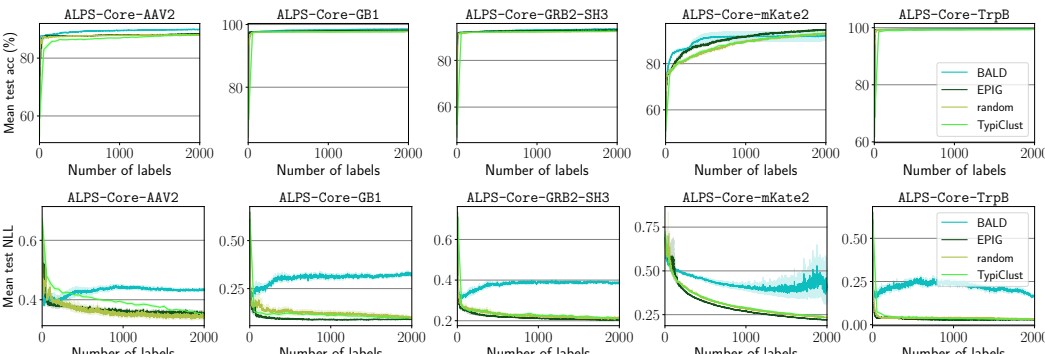

**Figure 30** Figure 2 with full vertical-axis ranges.

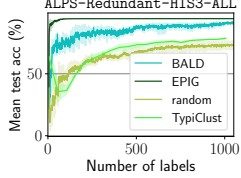
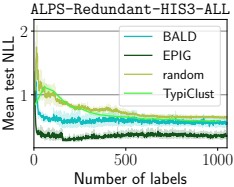

**Figure 31** Figure 4 with full vertical-axis ranges.

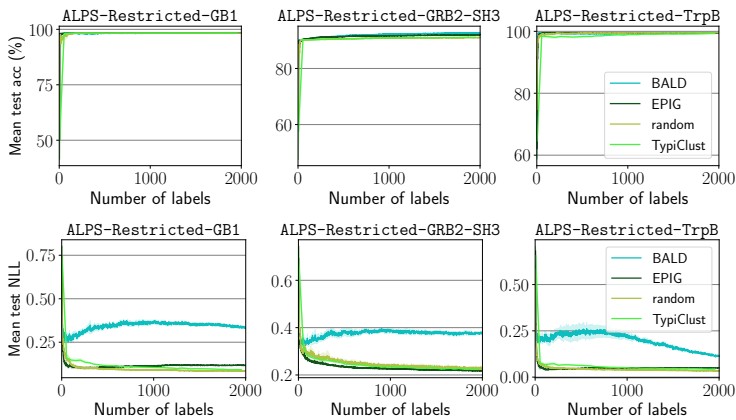

**Figure 32** Figure 5 with full vertical-axis ranges.

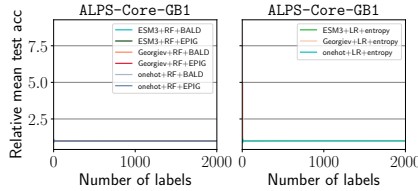 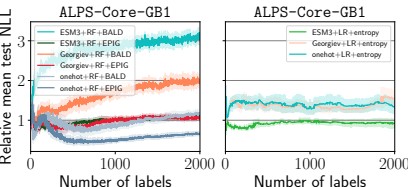

**Figure 33** Figure 7 with full vertical-axis ranges.

# E    LABEL NOISE

Here we provide some information regarding label noise in the `ALPS` problems, drawing from the papers introducing the original experimental datasets that we use to construct `ALPS`. We recommend referring to the full papers to better contextualise the content we quote.

**AAV2**    Bryant et al (2021) noted label noise "caused by low plasmid counts for specific variants"; discussed dealing with the noise by filtering based on plasmid count and by binarising measurements; and reported high correlations between experimental replicates (their Supplementary Figure 2).

**GB1**    Wu et al (2016) noted label noise for "10,639 missing variants (i.e. 6.6% of the sequence space) that had fewer than 10 sequencing read counts in the input library"; discussed dealing with the noise by filtering based on read count and imputing missing labels; and reported "high reproducibility in the data" and "fitness measurements. . . highly consistent with our previous study".

**GRB2**    Faure et al (2024) said they "obtained triplicate abundance measurements for 129,320 variants, which is 0.0007% of the sequence space" and the "measurements were highly reproducible".

**His3**    Pokusaeva et al (2019) said they "measured fitness for a total of 4,018,105 genotypes (875,151 unique amino acid sequences) with high accuracy" while noting that "For one segment, 9, the accuracy of our experiment was low"; they supported their judgements with an "accuracy analysis".

**mKate2**    Poelwijk et al (2019) used sequence barcoding and a Poisson noise model to "correct for mis-sorting events and unobserved spurious mutations that can introduce errors in assigning phenotypes", leading to "removal of 2% of counts, after which final enrichments were calculated".

**TrpB**    Johnston et al (2024) reported that their "fitness values of overlapping subsets of the 3- and 4- site libraries were highly correlated" and that "Analysis of the nearly one million unique codon combinations sampled showed that synonymous mutations had minimal impact on fitness", indicating their measurements of protein fitness had a good level of internal consistency.

