# OpenReview forum: "Improving Active-Learning Evaluation, with Applications to Protein-Property Prediction"
_ICLR.cc/2026/Conference — Submitted to ICLR 2026_

### Official Review · Reviewer_bve1 · 2025-10-19

**Soundness:** 3
**Presentation:** 3
**Contribution:** 3
**Rating:** 6
**Confidence:** 4

**Summary:**

This paper introduces a protein-property-based benchmark aimed at exposing weaknesses in current active learning evaluations that rely on overly curated datasets, offering more realistic yet domain-specific test conditions.

**Strengths:**

1.The paper raises a highly insightful and underexplored issue, that standard active learning benchmarks are often constructed from carefully curated datasets lacking truly non-informative samples. This leads to evaluations that may not reflect real-world conditions, where many samples are noisy, redundant, or irrelevant. The authors’ recognition of this gap and their effort to design ALPS as a more realistic, less curated benchmark represent a meaningful and original contribution to improving the validity of active learning evaluation.
2.Although the benchmark is built within a specific biological domain, the paper successfully designs several realistic and broadly relevant challenges such as label imbalance, redundant inputs, and acquisition restrictions. These conditions effectively test how active learning methods behave under real-world constraints. The protein-property setting provides a natural environment where such challenges can be observed and controlled, making ALPS a useful complement to existing benchmarks. It may not yet serve as a domain-independent standard, but it is a solid and practical step toward more reliable and diagnostic active learning evaluation.

**Weaknesses:**

1.While the paper insightfully identifies that existing active learning benchmarks rely on overly curated datasets, it does not quantitatively demonstrate that its proposed ALPS benchmark avoids the same issue. The authors only provide qualitative reasoning such as emphasizing that protein-property prediction data are experimentally derived and that ALPS minimizes filtering, but offer no measurable evidence (e.g., label noise level, data redundancy ratio, or informativeness variance) to substantiate the claim that ALPS is genuinely less curated. As a result, the paper does not fully address, in a quantifiable way, the very evaluation bias it aims to highlight.
2.The chosen experimental domain, protein-property prediction. is overly narrow to justify such a broad claim to me. The authors use this domain as evidence that AL evaluation should move away from curated datasets, yet this reasoning does not generalize: in many application areas, such as medical imaging or safety perception, highly curated test sets are necessary to ensure fairness and reliability rather than bias. The paper therefore conflates a domain-specific insight with a general methodological critique; without cross-domain evidence or domain-agnostic metrics, the claim that ALPS improves general AL evaluation remains under-supported.

**Questions:**

Could you provide some quantitative evidence or proxy metric (e.g., redundancy ratio, noise estimation, task-relevance measure) to support that the ALPS datasets are indeed less curated than standard benchmarks?

---

> ### Author Response · Authors · 2025-11-21
>
> Thank you for your review. We are pleased that you see the issues we identify as important, that you view our proposed set of active-learning problems as a natural and useful complement to existing problems. We aim to address your concerns below.
>
> ## **Degree of curation**
>
> > While the paper insightfully identifies that existing active learning benchmarks rely on overly curated datasets, it does not quantitatively demonstrate that its proposed ALPS benchmark avoids the same issue. The authors only provide qualitative reasoning such as emphasizing that protein-property prediction data are experimentally derived and that ALPS minimizes filtering, but offer no measurable evidence (e.g., label noise level, data redundancy ratio, or informativeness variance) to substantiate the claim that ALPS is genuinely less curated… Could you provide some quantitative evidence or proxy metric (e.g., redundancy ratio, noise estimation, task-relevance measure) to support that the ALPS datasets are indeed less curated than standard benchmarks?
>
> While much of what we care about is inherently qualitative, such as the concrete steps used to gather the datasets we are using, we appreciate wanting extra precision where possible. We have created a new Appendix E, in which we highlight some key information regarding label noise, drawing from the papers introducing the original experimental datasets. This complements the information in the original paper regarding data redundancy (eg, Appendix C.2.4). We do not believe informativeness variance is a well-defined quantity: a given label’s informativeness will often vary significantly from case to case, with the model at hand being a key determinant.
>
> ## **Focus on protein-property prediction**
>
> > The chosen experimental domain, protein-property prediction. is overly narrow to justify such a broad claim to me. The authors use this domain as evidence that AL evaluation should move away from curated datasets, yet this reasoning does not generalize: in many application areas, such as medical imaging or safety perception, highly curated test sets are necessary to ensure fairness and reliability rather than bias. The paper therefore conflates a domain-specific insight with a general methodological critique; without cross-domain evidence or domain-agnostic metrics, the claim that ALPS improves general AL evaluation remains under-supported.
>
> We firmly believe our focus on protein-property prediction is well justified. Evaluating active-learning methods on a new set of problems is not mutually exclusive with evaluating on existing problems, for example those based on image or text datasets: researchers can do both. Given this, the key priority in designing a new set of problems is to address the shortfalls in those problems that already exist. This is what we argue we have achieved with the ALPS problems. Doing so while incorporating a new data modality into the core active-learning literature provides further value: it is surely a strength to focus on a new domain if existing ones are well covered.

---

> > ### Comment · Reviewer_bve1 · 2025-11-22
> > **Thank you for your rebuttal**
> >
> > Thank you for your rebuttal and for addressing my concerns. After reading the other reviews, I agree with the general concern that this work, as framed, is closer to a domain-specific benchmark than a domain-general evaluation framework, and several of the broader claims are difficult to justify without stronger quantitative evidence or cross-domain validation.
> > I give positive rating to this work because I think the paper addresses a problem direction that I consider genuinely important, that is, much of the AL literature evaluates methods on clean, curated datasets and implicitly claims robustness to challenges such as noise, redundancy, or imbalance. In contrast, the authors focus on a domain where these challenges naturally exist and cannot be artificially removed. From this perspective, the work provides a meaningful contribution by grounding AL evaluation in a setting where the difficulties are real rather than simulated.

---

### Official Review · Reviewer_SoqQ · 2025-10-23

**Soundness:** 2
**Presentation:** 3
**Contribution:** 3
**Rating:** 4
**Confidence:** 4

**Summary:**

The authors posit that current Active Learning (AL) evaluations are done on unrealistic datasets and therefore skew the reported performance on acquisition functions. The main drawback of current datasets is the highly curated nature of the data samples, where even the most uninformative sample at any point is still well correlated with the target variable.
To remedy this situation, the authors propose ALPS, a protein-property prediction benchmark for AL, comprised of 6 protein-datasets that are evaluated on 3 novel tasks (in addition to standard batch AL).
In addition to describing the construction of each task, the authors mainly provide results for two Bayesian acquisition functions (BALD, EPIG) on their benchmark.

**Strengths:**

- Valuable collection of new and novel tasks for AL that significantly expand the types of problems AL is tested on
- Large bank of implemented encoder models, which should serve as a role-model for other benchmark papers in and across AL

**Weaknesses:**

- TypiClust is a bad representative of diversity-based methods for the tested budgets, as it was originally developed for ulta-low budgets and has shown lacking performance in other setups (e.g. Werner et al)
- The "Redundant" Task is fundamentally ill-posed: Constructing a machine learning task with 3 classes in the training set, but only 2 in the test set is scientifically not sound. This is mostly a problem with the description of the task. As an alternative, the authors should consider to stick to the original setup of (Bickford et al) that is defined on the basis of class imbalance. This would result in the following datasets (based on App. C.2.4): D_pool (86.8% neither, 12.4% Class 0, 0.7% Class 1); D_test (86.8% neither, 12.4% Class 0, 0.7% Class 1); D* (0% neither, 50% Class 0, 50% Class 1). All methods not taking advantage of D*, drastically fall behind EPIG in this setup.
- Missing technical details: (i) number of repetitions for each experiment, (ii) training setup (data augmentation, cold vs. warm-starts, seeding, etc.). Since this paper is a benchnark, we argue would that these details are especially important

**Questions:**

- Where are the results for BADGE, BAIT and ProbCover? The paper states in line 254 that these methods are implemented. All three methods represent strong acquisition functions in the field, so it is highly relevant, whether they exhibit the same shortcomings. Especially ProbCover is a way better fit to the tested budgets than TypiClust (see weakness as well).
- Some of your ablation studies - especially Fig. 3 - seem to indicate a structural problem in your experimental setup, as the behavior of the acquisition does not seem to correspond to the imbalance ratio. What number of repetitions was employed in the computation of your results and have you checked you inter-experiment variance? (see Werner et al for analysis on the number of repetitions needed for stable AL experiments)

[Brickford et al] Bickford Smith, Kirsch, Farquhar, Gal, Foster, & Rainforth (2023). Prediction-oriented Bayesian active learning. International Conference on Artificial Intelligence and Statistics. \
[Werner et al] Werner, Burchert, Stubbemann, & Schmidt-Thieme (2024). A cross-domain benchmark for active learning. Conference on Neural Information Processing Systems.

---

> ### Author Response · Authors · 2025-11-21
>
> Thank you for your review. We are pleased that you value the novel active-learning problems we introduce, and that you appreciate the range of models we provide in our codebase. We aim to address your concerns below.
>
> ## **TypiClust**
>
> > TypiClust is a bad representative of diversity-based methods for the tested budgets, as it was originally developed for ulta-low budgets and has shown lacking performance in other setups (e.g. Werner et al)
>
> You are right that TypiClust (Hacohen et al, 2022) was originally motivated by low label budgets. But we disagree that it is a bad representative of diversity-based methods for our experiments. Hacohen et al’s separation of low- and high-budget regimes does not have a universal numerical definition---they wrote that it “depends on the task and corresponding data distribution”---so there is no clear basis for claiming we are not in the low-budget regime. There are in fact multiple instances of TypiClust outperforming ProbCover with label budgets of the same order of magnitude as we consider here (Bae et al, 2024; Bickford Smith et al, 2024).
>
> ## **Construction of “Redundant” problem**
>
> > The "Redundant" Task is fundamentally ill-posed: Constructing a machine learning task with 3 classes in the training set, but only 2 in the test set is scientifically not sound. This is mostly a problem with the description of the task. As an alternative, the authors should consider to stick to the original setup of (Bickford et al) that is defined on the basis of class imbalance.
>
> We appreciate your attention to this. Note though that our setup here is actually aligned with the setup you cited as a good one (Bickford Smith et al, 2023): in their description of Redundant MNIST they wrote “the task involves classifying just images of 1s and 7s” and “we have three-way classification during training: 1 vs 7 vs neither”.
>
> ## **Experimental details**
>
> > Missing technical details: (i) number of repetitions for each experiment, (ii) training setup (data augmentation, cold vs. warm-starts, seeding, etc.). Since this paper is a benchnark, we argue would that these details are especially important
>
> We agree that these details are important. The original version of the paper reports the number of repetitions in multiple places, and also covers the aspects of the training setup that you mention, with the exception of data augmentation, since it is not used.
>
> ## **New results**
>
> > Where are the results for BADGE, BAIT and ProbCover? The paper states in line 254 that these methods are implemented. All three methods represent strong acquisition functions in the field, so it is highly relevant, whether they exhibit the same shortcomings. Especially ProbCover is a way better fit to the tested budgets than TypiClust (see weakness as well).
>
> We have added new BADGE and BAIT results to Appendix D.5 in the updated pdf. We would be happy to add them to the main content in the camera-ready paper. Note that they cannot simply be added to our existing plots in the main content because BADGE and BAIT are not directly applicable when using a random-forest prediction head, so our new results are based on alternative models with neural-network prediction heads.
>
> Given our discussion above, we actually believe our inclusion of TypiClust is well justified. We nevertheless appreciate your interest in seeing how well ProbCover performs. We will run additional experiments with ProbCover and would be happy to include the results in the camera-ready paper.
>
> ## **Class-imbalance experiment**
>
> > Some of your ablation studies \- especially Fig. 3 \- seem to indicate a structural problem in your experimental setup, as the behavior of the acquisition does not seem to correspond to the imbalance ratio. What number of repetitions was employed in the computation of your results and have you checked you inter-experiment variance?
>
> Thanks for checking this. For this experiment we currently have results for three repetitions, and the standard error is low (see the error bars in Figure 3). We would be happy to run more repetitions for the camera-ready paper.
>
> ---
>
> Bae et al (2024). Generalized coverage for more robust low-budget active learning. ECCV.
>
> Bickford Smith et al (2023). Prediction-oriented Bayesian active learning. AISTATS.
>
> Bickford Smith et al (2024). Making better use of unlabelled data in Bayesian active learning. AISTATS.
>
> Hacohen et al (2022). Active learning on a budget: opposite strategies suit high and low budgets. ICML.
>
> Pourahmadi et al (2021). A simple baseline for low-budget active learning. arXiv.

---

> > ### Comment · Reviewer_SoqQ · 2025-11-24
> >
> > Thank you for your response, we will be picking up the discussed points again:
> > ### TypiClust
> > Your reported sources for TypiClust performing on par with ProbCover are debatable, as (Bae et al, 2024) also test in the ultra-low budget regions (indicated by random outperforming uncertainty sampling and overall low accuracy values) and (Bickford Smith et al, 2024) do not report on TypiCust at all. \
> > However, you have a point with your remark about how to define "low-budget". Your performance curves in Fig. 2 saturate in accuracy towards the end indicating a converging classifier, which makes this distinctively not a low-budget setting. \
> > We would like to honor the efforts from the authors, providing additional results in App. D5 for BADGE and BAIT (which represent the SOTA in AL) and drop this weakness.
> >
> > ### Experimental Details
> > We missed the number of repetitions, that is our bad. \
> > However, your reporting is about the other technical details is very ambiguous:
> > 1. Do you use a global seed for your five repetitions, or do you use separate seeds (e.g. for data splitting and model initialization). Assuming, you use a global seed, how do you justify the vastly different starting positions for your tested methods (Fig. 2)?
> > 2. When you say "the model p is updated on D_train, does that mean you retain the old model weights, or do you reset the model and train it from scratch?
> >
> > ### Class Imbalance
> > We agree that your work and (Bickford Smith et al, 2024) are close to equivalent.
> > The difference is, that your description of the task (line 217-222) is just plain wrong in our opinion.
> > Saying that you have 3 classes during training and 2 classes during test makes it completely unclear, how the prediction head of any given model is supposed to look like (it can either have 2 or 3 classes, not both). Brickford et al circumvent this by fixing themselves on 3 classes, but with different distributions between train and test.
> >
> > ### Class-imbalance experiment
> > If you would like to argue that 3 repetitions of this experiment is enough, how do you explain the behavior that we observe in Fig. 3? Even if we don't expect some sort of linear correlation between Imbalance Ratio and Acc/NLL, we strongly expect the effect to be monotonic. \
> > We strongly urge the authors to compute another 3 repetitions with different seeds of this experiment and create a separate copy of Fig 3. This is the most reliable way of checking the inter-experiment variance that is in the system.
> >
> > \
> > Finally, we acknowledge the response of the authors to our concerns. \
> > If the authors manage to provide a convincing rebuttal to both of our last points ("Class Imbalance" and "Class-imbalance experiment), we will raise the score to 5.

---

> > > ### Author Response · Authors · 2025-11-29
> > >
> > > We are grateful for your continued engagement and for your offer to raise your score given a convincing response.
> > >
> > > ## **TypiClust**
> > >
> > > > We would like to honor the efforts from the authors, providing additional results in App. D5 for BADGE and BAIT (which represent the SOTA in AL) and drop this weakness.
> > >
> > > This is much appreciated.
> > >
> > > ## **Experimental details**
> > >
> > > > Do you use a global seed for your five repetitions, or do you use separate seeds (e.g. for data splitting and model initialization). Assuming, you use a global seed, how do you justify the vastly different starting positions for your tested methods (Fig. 2)?
> > >
> > > Thanks for checking this. We use a global seed. The difference in starting positions in our original plots was a plotting artefact that we have now addressed in the updated pdf. Because performance can be very poor at the start before quickly improving, naively plotting performance can lead to hard-to-read plots in which the gaps between curves are barely visible (see the plots in Appendix D.7 in the updated pdf). Aiming to improve readability, in our original plots we showed curves starting after a short “burn-in” period, with the difference in starting points resulting from different amounts of performance improvement in the burn-in period between methods. We recognise that this was confusing. In our new plots we show curves starting from the very beginning; in the main paper we use vertical axes with reduced ranges so that the gaps between curves are easier to see; and in Appendix D.7 we use full vertical-axis ranges for completeness.
> > >
> > > > When you say "the model p is updated on D\_train, does that mean you retain the old model weights, or do you reset the model and train it from scratch?
> > >
> > > Good question. Updating here means retraining from scratch. We will make this clearer in the paper.
> > >
> > > ## **Class imbalance**
> > >
> > > > Saying that you have 3 classes during training and 2 classes during test makes it completely unclear, how the prediction head of any given model is supposed to look like (it can either have 2 or 3 classes, not both). Brickford et al circumvent this by fixing themselves on 3 classes, but with different distributions between train and test.
> > >
> > > We can see how this was unclear. Our setup matches that used by Bickford Smith et al (2024). We have updated our wording in the paper with the aim of reducing confusion.
> > >
> > > ## **Class-imbalance experiment**
> > >
> > > > We strongly urge the authors to compute another 3 repetitions with different seeds of this experiment and create a separate copy of Fig 3\. This is the most reliable way of checking the inter-experiment variance that is in the system.
> > >
> > > There was actually a mistake on our side here---sorry. In our response we said Figure 3 showed results for three seeds, but we later checked this and found it showed results for five seeds. This means we can be more confident in the results we have. We recognise your request for extra seeds though, and we are currently running with more.
> > >
> > > > If you would like to argue that 3 repetitions of this experiment is enough, how do you explain the behavior that we observe in Fig. 3? Even if we don't expect some sort of linear correlation between Imbalance Ratio and Acc/NLL, we strongly expect the effect to be monotonic.
> > >
> > > This is a good question that aligns with a comment by Reviewer hbF9: “in the cases where thresholds were used to create class imbalance, I wonder if the performance of the tested methods were influenced by the imbalance, or the label transformation”. The interaction of these factors might explain why the effect of the imbalance ratio is not monotonic. We will run additional experiments to provide some clarity.

---

> > > > ### Author Response · Authors · 2025-12-02
> > > >
> > > > Figure 3 now shows results for six seeds, taking us up to the total number suggested in your comment. We will add more once ongoing runs have finished.

---

### Official Review · Reviewer_hbF9 · 2025-10-30

**Soundness:** 3
**Presentation:** 3
**Contribution:** 3
**Rating:** 4
**Confidence:** 4

**Summary:**

This paper introduces Active Learning on Protein Sequences (ALPS), a set of active learning benchmarks that are designed to incorporate challenges that are prevalent in real-world settings. These benchmarks can help to address issues that active learning methods may face in practical use cases, issues which might otherwise go unnoticed if the methods are usually tested on better curated datasets. The paper also presents experiment results of testing active learning methods on the ALPS dataset, the results demonstrate that some popular approaches (such as BALD and EPIG) perform suboptimally under certain conditions and exhibit issues such as uncertainty miscalibration, sensitivity to class imbalance etc. Overall, the new benchmark would allow for a more robust evaluation and can help drive the development of improved active learning methods.

**Strengths:**

- The paper introduces a novel benchmark that addresses the limitation of existing heavily curated benchmarks datasets by including challenges commonly encountered in real-world settings. This allows for more realistic and robust evaluation of AL methods for practical applications, and empirical experiments demonstrate how it can reveal previously unknown failure modes in popular AL methods.
- The empirical experiments are very thorough, evaluating multiple encoders, prediction heads, acquisition methods etc. across multiple tasks. Components appear to be implemented in a modular setup which allows for ease of testing specific parts of the AL pipeline.

**Weaknesses:**

- There is a lack of data quality evaluation for this benchmark dataset. There is little discussion and evaluation regarding how reliable the labels are, or the level of ambiguity / noise in the dataset. It is crucial for benchmark datasets to ensure the quality of its data and labels, hence a more thorough evaluation of the reliability of the data would strengthen the benchmark’s credibility and utility.
- While the benchmark supports a large array of encoders / predictors / acquisition methods, the main evaluations are limited (focused mainly on BALD, EPIG, TypiClust and random), resulting in many other popular AL methods not being fully explored and making it harder to draw conclusions about how the current methods behave in this more challenging, real-world-like setting
- Protein sequencing is a highly domain specific task, and it is unclear whether the observed challenges that various AL methods face in this benchmark would transfer to other tasks of modalities. This potentially limits the generalizability of the insights gained from this benchmark.

**Questions:**

- Some tasks in this benchmark were converted from regression tasks into a binary classification tasks using constant thresholds. I am curious if the authors have any insight into if this binarization affects the quality of the data (might introduce ambiguity around the decision boundary). In particular, in the cases where thresholds were used to create class imbalance, I wonder if the performance of the tested methods were influenced by the imbalance, or the label transformation. Could the authors isolate the two contributing factors here to better understand the consequences of each change?
- Could the authors provide more information / rationale behind their dataset design decisions? For example, why were specific imbalance ratios or batch sizes chosen, and how these might have an impact on the observed behaviours of the tested methods.

---

> ### Author Response · Authors · 2025-11-21
>
> Thank you for your review. We are pleased that you see our work as addressing existing issues and allowing more realistic evaluations, and that you found our experiments thorough. We aim to address your concerns below.
>
> ## **Label noise**
>
> > There is little discussion and evaluation regarding how reliable the labels are, or the level of ambiguity / noise in the dataset.
>
> Great point. We have created a new Appendix E, in which we highlight some key information regarding label noise, drawing from the papers introducing the original experimental datasets.
>
> ## **New results**
>
> > the main evaluations are limited (focused mainly on BALD, EPIG, TypiClust and random), resulting in many other popular AL methods not being fully explored and making it harder to draw conclusions about how the current methods behave in this more challenging, real-world-like setting
>
> We have added new BADGE and BAIT results to Appendix D.5 in the updated pdf. We would be happy to add them to the main content in the camera-ready paper. Note that they cannot simply be added to our existing plots in the main content because BADGE and BAIT are not directly applicable when using a random-forest prediction head, so our new results are based on alternative models with neural-network prediction heads.
>
> ## **Generalisability**
>
> > Protein sequencing is a highly domain specific task, and it is unclear whether the observed challenges that various AL methods face in this benchmark would transfer to other tasks of modalities. This potentially limits the generalizability of the insights gained from this benchmark.
>
> You are right to think about the generalisability of insights from a given evaluation; this is a universal challenge in machine-learning evaluation. But we actually think this is a point in favour of our work, not against it. If we are interested in using active learning for applications in biochemistry and drug design, and if we are wary of generalising too strongly from a given evaluation, it is a problem that the active-learning community relies so strongly on problems that are so detached from these applications.
>
> ## **Label binarisation**
>
> > in the cases where thresholds were used to create class imbalance, I wonder if the performance of the tested methods were influenced by the imbalance, or the label transformation. Could the authors isolate the two contributing factors here to better understand the consequences of each change?
>
> Good question. We will run additional experiments to disentangle these two factors. We would be happy to add the results to the camera-ready paper.
>
> ## **Imbalance ratios and batch sizes**
>
> > why were specific imbalance ratios or batch sizes chosen, and how these might have an impact on the observed behaviours of the tested methods
>
> The imbalance ratios and batch sizes were just chosen with an aim to cover a useful range of scales with the computational resources we had available. We would be happy to include additional configurations in further experiments for the camera-ready paper if you see value in that.

---

### Official Review · Reviewer_6sSo · 2025-11-01

**Soundness:** 3
**Presentation:** 3
**Contribution:** 2
**Rating:** 2
**Confidence:** 4

**Summary:**

The paper argues that many active learning evaluations use heavily curated datasets and therefore under-stress data acquisition. It proposes a new benchmark suite derived from experimental protein datasets, ALPS to better reflect messy pools, task adaptation, class imbalance, acquisition restrictions, and batching. Using ALPS, the authors benchmark several acquisition strategies with pretrained protein encoders and various heads.

**Strengths:**

1. The application of active learning to protein-property prediction is both novel and important. This work commendably addresses a challenging and under-explored domain, moving beyond the standard.

2. Clear formalization of evaluation goals via frequentist risk and explicit factors (task, loss, pool, p_train, p_eval, model, budgets).

3. Codebase and datasets are provided for reproducibility. This represents a contribution to the community, facilitates reproducibility, and allows for future extensions.

**Weaknesses:**

1. The empirical evaluation is not comprehensive. The authors state that the codebase implements 12 acquisition methods, including widely-used gradient-based methods like BADGE and BAIT. However, the core experimental analysis (Section 6) focuses almost exclusively on BALD and EPIG, with TypiClust and random sampling as baselines. Given that BADGE, in particular, is a strong and practical baseline in many deep AL settings, its omission from the main comparisons (Figures 2-8) is a significant flaw and undermines the comprehensiveness of the benchmark.

2. A central motivation of the paper is to improve practical evaluation, highlighting the cost of labeling proteins. However, the analysis of "cost" is limited to the number of acquired labels. It completely omits the computational cost (e.g., wall-clock time, FLOPs, or memory footprint) of the acquisition step itself. Since the search space of protein is prohibitively large, and the prediction of the properties is also expensive, it is necessary to discuss about the computational cost in my view. Methods like EPIG, BALD, and BADGE have vastly different computational overheads. To truly evaluate practical utility, the benchmarks must present performance as a function of both label cost and computational cost.

3. The reliance on binarization as the primary task formulation is a limitation. While the authors investigate varying thresholds in the ALPS-Unbalanced tasks, the binarization in the ALPS-Core tasks (based on a single wild-type reference value) can be arbitrary and may not reflect the true scientific objective (e.g., finding a variant with maximum fluorescence). The paper would be stronger if it either (a) included a sensitivity analysis for this threshold choice in the Core tasks or (b) directly evaluated performance on the native regression task, which the paper acknowledges but does not explore.

4. I think the motivation is questionable. On one hand, many existing AL benchmarks are also derived from real world, and contains useless even harmful data; on the other hand, ALPS is still a retrospective simulation on a specific and static pool, some of its crafted variants, e.g., ALPS-Unbalanced, ALPS-Redundant, may not really reflect the real scenarios. A truly real-world setting, particularly in protein engineering or drug discovery, is often an optimization or search problem (e.g., "find the protein with the highest binding affinity") rather than a pool-based classification task. This objective is better modeled by Bayesian Optimization or an exploration-exploitation framework. Therefore, I think the problem of active learning evaluation has not been sufficiently addressed.

5. The paper's positioning and literature review almost entirely overlook the substantial, existing body of work in active learning for drug discovery and virtual screening. This field has its own established benchmarks, models, and practical settings and has extensively studied the exact problems the authors claim to motivate their work. The paper should not be comparing itself only to generic CV/NLP benchmarks but to the established practices in its own target domain.

**Questions:**

Please see weakness.

---

> ### Author Response · Authors · 2025-11-21
>
> Thank you for your review. We are pleased that you see protein-property prediction as an important domain to focus on, that you found our formalisation of active-learning evaluation clear, and that you appreciate our efforts to enable reproducibility and extensibility. We aim to address your concerns below.
>
> ## **New results**
>
> > Given that BADGE, in particular, is a strong and practical baseline in many deep AL settings, its omission from the main comparisons (Figures 2-8) is a significant flaw
>
> We have added new BADGE and BAIT results to Appendix D.5 in the updated pdf. We would be happy to add them to the main content in the camera-ready paper. Note that they cannot simply be added to our existing plots in the main content because BADGE and BAIT are not directly applicable when using a random-forest prediction head, so our new results are based on alternative models with neural-network prediction heads.
>
> ## **Computational cost**
>
> > the analysis of "cost" is limited to the number of acquired labels. It completely omits the computational cost (e.g., wall-clock time, FLOPs, or memory footprint) of the acquisition step itself.
>
> Thanks for flagging this. We have included wall-clock times indicating relative computational costs in Appendix D.6 in the updated pdf.
>
> ## **Label binarisation**
>
> > the binarization in the ALPS-Core tasks (based on a single wild-type reference value) can be arbitrary and may not reflect the true scientific objective (e.g., finding a variant with maximum fluorescence).
>
> We contest the idea that the binarisation we use is arbitrary. We use wildtype property values as thresholds---as also done by, for example, Notin et al (2023)---precisely to avoid this. Then the classification task is predicting whether a given mutant protein has a higher or lower property value than the wildtype protein, which aligns with how the original experimental data was collected.
>
> While we agree that there are cases where predicting a continuous value would be relevant, there are many where classification is relevant. For example, as we will touch on below regarding your question of motivation, the property of interest might be serving as a constraint, not an objective, in optimisation.
>
> As you note, our ALPS-Unbalanced problems depart from the standard wildtype-based binarisation for a focused exploration of class imbalance. We believe these problems serve their purpose well.
>
> ## **Motivation**
>
> > many existing AL benchmarks are also derived from real world, and contains useless even harmful data
>
> We would be happy to comment on any specific benchmarks you have in mind.
>
> > ALPS is still a retrospective simulation on a specific and static pool
>
> You are right to point out that non-pool-based problems are possible. But we disagree with the implication that our use of pools is a substantive limitation of this work. The vast majority of the active-learning literature focuses on pool-based acquisition (Settles, 2012), which makes sense as a practical approach given the difficulty of non-pool-based acquisition (Lang & Baum, 1992). Given a lack of well-established methods for non-pool-based acquisition, we strongly believe it is premature to criticise active-learning evaluations for using pools.
>
> > A truly real-world setting, particularly in protein engineering or drug discovery, is often an optimization or search problem (e.g., "find the protein with the highest binding affinity") rather than a pool-based classification task
>
> We agree that some applications are optimisation problems. But that is not mutually exclusive with active learning being the right approach for other applications. As we explain under “Applications” in Section 4, in protein engineering we might want a predictive model to serve as one component in a downstream optimisation objective or as a constraint within an optimisation problem, and in basic science we might want a predictive model for the insights it gives into a biological phenomenon like epistasis.
>
> ## **Existing work on applied active learning**
>
> > The paper's positioning and literature review almost entirely overlook the substantial, existing body of work in active learning for drug discovery and virtual screening. This field has its own established benchmarks, models, and practical settings and has extensively studied the exact problems the authors claim to motivate their work. The paper should not be comparing itself only to generic CV/NLP benchmarks
>
> Again we would be happy to comment on any specific benchmarks you have in mind, while noting that our paper is directly addressing the core active-learning-methods community, so the comparison to existing CV- and NLP-heavy benchmarks within that community is pertinent.
>
> ---
>
> Lang & Baum (1992). Query learning can work poorly when a human oracle is used. IJCNN.
>
> Notin et al (2023). ProteinGym: large-scale benchmarks for protein design and fitness prediction. NeurIPS.
>
> Settles (2012). Active Learning. Morgan & Claypool.

---

### Meta-Review · Area_Chair_zvpP · 2025-12-14

**Summary:**

This paper introduces ALPS (Active Learning on Protein Sequences), a benchmark suite designed to highlight limitations of existing active learning (AL) evaluations that rely heavily on overly curated datasets. By constructing several challenging task variants (e.g., unbalanced and redundant settings) from experimentally derived protein-property datasets, the authors aim to better approximate real-world data acquisition scenarios and systematically analyze the behavior of commonly used AL acquisition strategies. The release of an open-source codebase further enhances the potential long-term value of this work by enabling extensibility and future benchmarking efforts.

Despite the strong and well-articulated motivation, the reviewers identified several substantial concerns that limit the current strength of the submission. These include an incomplete empirical evaluation with key acquisition baselines missing from the main comparisons, the absence of an analysis of computational cost despite its importance in realistic protein search settings, and issues related to task formulation—particularly the reliance on binarization and threshold-based label construction. Additional concerns were raised regarding experimental design clarity, benchmark validity, and the generality of the conclusions drawn.

The authors have made efforts during the rebuttal phase to address some of these issues by adding supplementary experiments and clarifications, which partially mitigates several reviewer concerns. However, in the current version, important questions regarding evaluation completeness, experimental rigor, and problem formulation remain insufficiently resolved. Further refinement and clearer presentation would be necessary for the benchmark to fully support the paper’s central claims.

Overall, while the work is promising and addresses an important problem, I recommend rejection in its current version.

**Reviewer Concerns:**

Reviewer 6sSo raised concerns about the practical relevance and completeness of the evaluation, noting that key acquisition baselines such as BADGE were initially missing and that the analysis focused solely on labeling cost while ignoring computational cost, which is critical in large protein search spaces. The reviewer also questioned the reliance on binarized classification tasks, arguing that threshold-based labels may be arbitrary and misaligned with real scientific objectives, and further suggested that many protein engineering problems are better framed as optimization or search tasks rather than pool-based active learning. Additionally, concerns were raised regarding limited engagement with prior work in drug discovery and virtual screening.

In the rebuttal, the authors added results on BADGE and BAIT, and provided additional discussion on computational cost, label binarization, motivation, and related work. While these additions improve clarity and partially address the reviewer’s concerns, they do not fully resolve the underlying questions regarding evaluation completeness and the realism of the problem formulation.

----

Reviewer hbF9 raised concerns regarding data quality, evaluation coverage, and generalizability. Specifically, the reviewer noted the lack of systematic analysis of label reliability, noise, and ambiguity, the limited exploration of acquisition methods despite broad claimed support, and questioned whether insights drawn from protein sequence tasks generalize to other domains or modalities. Additional concerns were raised about the use of threshold-based binarization, particularly in imbalanced settings, and about insufficient justification for dataset design choices, such as imbalance ratios and batch sizes.

In the rebuttal, the authors provided additional discussion and results on label noise, included new experimental results, and expanded their responses on label binarization, generalizability, and dataset design. While these additions partially address the reviewer’s concerns, the analysis of imbalance ratios and batch sizes remains incomplete, and the discussion of generalizability beyond protein sequencing is still limited, leaving some key questions insufficiently resolved.

---

Reviewer SoqQ raised concerns regarding experimental design, methodological choices, and missing technical details. Specifically, the reviewer argued that TypiClust is not a suitable representative of diversity-based methods under the tested labeling budgets, and suggested alternatives such as ProbCover. They also questioned the scientific validity and clarity of the Redundant task design, and emphasized the absence of essential experimental details, including the number of repetitions, variance across runs, seeding strategies, and training protocols. In addition, the reviewer asked why results for implemented methods such as BADGE, BAIT, and ProbCover were not reported, and whether some observed behaviors could be attributed to insufficient repetitions or high variance.

In the rebuttal, the authors provided structured and substantive responses, clarified the experimental setup, and addressed several of the methodological concerns. Overall, these responses partially mitigate the reviewer’s concerns.

---

Reviewer bve1 questioned whether the paper quantitatively substantiates its central claim that ALPS avoids the over-curation issues present in existing active learning benchmarks, noting that the original submission relied primarily on qualitative arguments without measurable evidence. The reviewer also expressed concern that conclusions drawn from a single domain—protein-property prediction—were used to support broader claims about active learning evaluation, which may not generalize to domains where curated datasets are necessary.

During the rebuttal, the authors provided clearer quantitative justification and clarification of scope, and explicitly acknowledged the domain-specific nature of their claims. These responses were well received and largely resolved the reviewer’s concerns, resulting in a positive assessment.

**Reviewer Scores:**

Based on the rebuttal and subsequent discussion, Reviewer 6sSo and Reviewer hbF9 are unlikely to substantially change their original scores, as several of their core concerns remain only partially addressed. Reviewer bve1 responded positively to the rebuttal and may increase their score accordingly. Reviewer SoqQ is expected to maintain their original score.

---

### Decision · Program_Chairs · 2026-01-26

Reject